

Earth System
Dynamics

# Projecting Antarctica's contribution to future sea level rise from basal ice shelf melt using linear response functions of 16 ice sheet models (LARMIP-2)

Anders Levermann[1,2,3], Ricarda Winkelmann[1,3], Torsten Albrecht[1], Heiko Goelzer[4,5],
Nicholas R. Golledge[6,7], Ralf Greve[8], Philippe Huybrechts[9], Jim Jordan[10], Gunter Leguy[11],
Daniel Martin[12], Mathieu Morlighem[13], Frank Pattyn[5], David Pollard[14], Aurelien Quiquet[15],
Christian Rodehacke[16,17], Helene Seroussi[18], Johannes Sutter[17,19], Tong Zhang[20],
Jonas Van Breedam TS1[9], Robert DeConto[21], Christophe Dumas[15], Julius Garbe[1,3],
G. Hilmar Gudmundsson[10], Matthew J. Hoffman[20], Angelika Humbert[17,22], Thomas Kleiner[17],
William H. Lipscomb[11], Malte Meinshausen[23,1], Esmond Ng[12], Sophie M. J. Nowicki[24], Mauro Perego[25],
Stephen F. Price[20], Fuyuki Saito[26], Nicole-Jeanne Schlegel[18], Sainan Sun[5], and
Roderik S. W. van de Wal[4,27]

[1]Potsdam Institute for Climate Impact Research, Potsdam, Germany
[2]LDEO, Columbia University, New York, USA
[3]Institute of Physics and Astronomy, University of Potsdam, 14476 Potsdam, Germany
[4]Institute for Marine and Atmospheric research Utrecht, Utrecht University, Utrecht, the Netherlands
[5]Laboratoire de Glaciologie, Université libre de Bruxelles (ULB), Brussels, Belgium
[6]Antarctic Research Centre, Victoria University of Wellington, Wellington 6140, New Zealand
[7]GNS Science, Avalon, Lower Hutt 5011, New Zealand
[8]Institute of Low Temperature Science, Hokkaido University, Sapporo 060-0819, Japan
[9]Earth System Science & Departement Geografie, Vrije Universiteit Brussel, Brussels, Belgium
[10]Department of Geography and Environmental Sciences, University of Northumbria, Newcastle, UK
[11]Climate and Global Dynamics Laboratory, National Center for Atmospheric Research, Boulder, CO, USA
[12]Lawrence Berkeley National Laboratory, Berkeley, CA, USA
[13]Department of Earth System Science, University of California Irvine, Irvine, CA, USA
[14]Earth and Environmental Systems Institute, Pennsylvania State University,
University Park, Pennsylvania, USA
[15]Laboratoire des Sciences du Climat et de l'Environnement, CEA/CNRS-INSU/UVSQ,
Gif-sur-Yvette CEDEX, France
[16]Danish Meteorological Institute, Arctic and Climate, Copenhagen, Denmark
[17]Alfred Wegener Institute Helmholtz Centre for Polar and Marine Research, Bremerhaven, Germany
[18]Jet Propulsion Laboratory, California Institute of Technology, Pasadena, CA, USA
[19]Physics Institute, University of Bern, Bern, Switzerland
[20]Theoretical Division, Los Alamos National Laboratory, Los Alamos, New Mexico 87545, USA
[21]Department of Geosciences, University of Massachusetts, Amherst, Massachusetts, USA
[22]Department of Geosciences, University of Bremen, Bremen, Germany
[23]Climate & Energy College, School of Earth Sciences, University of Melbourne, Parkville, Victoria, Australia
[24]NASA Goddard Space Flight Center, Greenbelt, MD, USA
[25]Center for Computing Research, Sandia National Laboratories, Albuquerque, New Mexico 87185, USA
[26]Japan Agency for Marine-Earth Science and Technology, Yokohama, Japan
[27]Geosciences, Physical Geography, Utrecht University, Utrecht, the Netherlands

**Correspondence:** Anders Levermann (anders.levermann@pik-potsdam.de)

Published by Copernicus Publications on behalf of the European Geosciences Union.

Please note the remarks at the end of the manuscript.

Received: 9 May 2019 – Discussion started: 23 May 2019
Revised: 26 November 2019 – Accepted: 4 December 2019 – Published:

**Abstract.** The sea level contribution of the Antarctic ice sheet constitutes a large uncertainty in future sea level projections. Here we apply a linear response theory approach to 16 state-of-the-art ice sheet models to estimate the Antarctic ice sheet contribution from basal ice shelf melting within the 21st century. The purpose of this computation is to estimate the uncertainty of Antarctica's future contribution to global sea level rise that arises from large uncertainty in the oceanic forcing and the associated ice self melting. Ice shelf melting is considered to be a major if not the largest perturbation of the ice sheet's flow into the ocean. However, by computing only the sea level contribution in response to ice shelf melting, our study is neglecting a number of processes such as surface-mass-balance-related contributions. In assuming linear response theory, we are able to capture complex temporal responses of the ice sheets, but we neglect any self-dampening or self-amplifying processes. This is particularly relevant in situations in which an instability is dominating the ice loss. The results obtained here are thus relevant, in particular wherever the ice loss is dominated by the forcing as opposed to an internal instability, for example in strong ocean warming scenarios. In order to allow for comparison the methodology was chosen to be exactly the same as in an earlier study (Levermann et al., 2014) but with 16 instead of 5 ice sheet models. We include uncertainty in the atmospheric warming response to carbon emissions (full range of CMIP5 climate model sensitivities), uncertainty in the oceanic transport to the Southern Ocean (obtained from the time-delayed and scaled oceanic subsurface warming in CMIP5 models in relation to the global mean surface warming), and the observed range of responses of basal ice shelf melting to oceanic warming outside the ice shelf cavity. This uncertainty in basal ice shelf melting is then convoluted with the linear response functions of each of the 16 ice sheet models to obtain the ice flow response to the individual global warming path. The model median for the observational period from 1992 to 2017 of the ice loss due to basal ice shelf melting is 10.2 mm, with a likely range between 5.2 and 21.3 mm. For the same period the Antarctic ice sheet lost mass equivalent to 7.4 mm of global sea level rise, with a standard deviation of 3.7 mm (Shepherd et al., 2018) including all processes, especially surface-mass-balance changes. For the unabated warming path, Representative Concentration Pathway 8.5 (RCP8.5), we obtain a median contribution of the Antarctic ice sheet to global mean sea level rise from basal ice shelf melting within the 21st century of 18 cm, with a likely range (66th percentile around the mean) between 9 and 38 cm and a very likely range (90th percentile around the mean) between 6 and 61 cm TS2. For the RCP2.6 warming path, which will keep the global mean temperature below 2 °C of global warming and is thus consistent with the Paris Climate Agreement, the procedure yields a median of 14 cm of global mean sea level contribution. The likely range for the RCP2.6 scenario is between 7 and 27 cm, and the very likely range is between 5 and 40 cm TS3. The structural uncertainties in the method do not allow for an interpretation of any higher uncertainty percentiles. We provide projections for the five Antarctic regions and for each model and each scenario separately. The rate of sea level contribution is highest under the RCP8.5 scenario. The maximum within the 21st century of the median value is 4 cm per decade, with a likely range between 2 and 9 cm per decade and a very likely range between 1 and 14 cm per decade.

## 1   Introduction

The Antarctic ice sheet has been losing mass at an increasing rate over the past decades (Rignot et al., 2019; Shepherd et al., 2018). Projections of changes in ice loss from Antarctica still constitute the largest uncertainty in future sea level projections (Bamber et al., 2019; Bamber and Aspinall, 2013; Church et al., 2013; Schlegel et al., 2018; Slangen et al., 2016). Evidence from paleorecords and regional and global climate models suggests that snowfall onto Antarctica follows a relation similar to the Clausius–Clapeyron law (Clapeyron, 1834; Clausius, 1850) of an increase by about 6 % for every degree of global warming (Frieler et al., 2015; Lenaerts et al., 2016; Medley and Thomas, 2019; O'Gorman

et al., 2012; Palerme et al., 2014, 2017; Previdi and Polvani, 2016). The current snowfall onto Antarctica is of the order of $8 \, \mathrm{mm \, yr^{-1}}$ in global sea level equivalent; i.e. an increase in snowfall will decrease global sea level of the order of half a millimetre for every degree of warming (van de Berg et al., 2006; Lenaerts et al., 2012). Surface melting is likely to play a minor role as a direct ice loss mechanism within the 21st century, but it might initiate other ice loss processes such as hydrofracturing and subsequent cliff calving with the potential for much higher ice loss than any other process (DeConto and Pollard, 2016; Pollard and DeConto, 2009). An important process of additional ice loss from Antarctica is basal ice shelf melt and the associated acceleration of ice flow across the grounding line (Bindschadler et al., 2013; Jenkins et al.,

2018; Nowicki et al., 2013, 2016; Reese et al., 2018a; Rignot et al., 2013; Shepherd et al., 2004).

Here we follow a very specific procedure that is designed to estimate the uncertainty of future ice loss from Antarctica as it can be induced by basal ice shelf melting. We follow exactly the same procedure as in Levermann et al. (2014) but with 16 ice sheet models instead of 3 models with a dynamic representation of ice shelves (although five models participated in the earlier study, only three of them had a dynamic representation of ice shelves). At the core of the approach is a linear response theory (Good et al., 2011; Winkelmann and Levermann, 2013), which is explained together with the models used in more detail in Sect. 2. The ice sheet models used here all take part in the initMIP intercomparison project for Antarctica (Seroussi et al., 2019) within the overall IS-MIP6 initiative (Goelzer et al., 2018; Nowicki et al., 2016). Section 3 provides the hindcasting for the observational period and Sect. 4 gives the results of the computation for the 21st century. The last section provides conclusions and discussions. Although we will not repeat details of the method in all aspects and refer to the earlier publication for that, we will summarize it in Sect. 2 in order to provide a paper that is understandable on its own. A detailed analysis as to why the 16 different models respond differently cannot be provided in this publication due to both space limitations and the fact that each of these analyses would constitute a full-scale publication in itself. We provide a synthesis of the results and refer to potential future studies by the individual modelling groups for details on the individual model results.

The purpose of this study is to estimate the uncertainty of basal-melt-induced sea level contribution from Antarctica as it is caused by the uncertainty in the basal melt forcing. While ice shelf melting is considered to be a major if not the largest perturbation of the ice sheet's flow into the ocean, the approach neglects a number of processes such as surface-mass-balance-related contributions (Bamber et al., 2018; Rignot et al., 2019) and their feedbacks (Levermann and Winkelmann, 2016). In assuming linear response theory, we are able to capture complex temporal responses of the ice sheets, but we neglect any self-dampening or self-amplifying processes. This is particularly relevant in situations in which an instability is dominating the ice loss. The results obtained here are thus relevant, in particular wherever the ice loss is dominated by the forcing as opposed to an internal instability, for example in strong warming scenarios.

In contrast to the study here, individual model simulations with specific time series of basal ice shelf forcing for a specific ice sheet model can be better used to understand specific processes and yield much more precise results for this specific basal melt forcing. The main contribution of this study is the investigation of the response of the models to the full range of uncertain forcing and the combination of this for all the different ice sheet models. In addition, the switch-on experiments at the basis of the analysis allow for a comparison of the different model responses to a very simple and generic forcing and might be used to improve the models or at least know how one specific model compares to the others in a specific region.

It is important to note that in this study no changes in the surface mass balance are taken into account, nor are any other ice loss processes other than the ice dynamic discharge into the ocean as it is induced from an increase in basal ice shelf melting. The term "Antarctic contribution to sea level rise" is used in this study to refer to the sea-level-relevant ice loss induced from basal ice shelf melting only.

## 2 Projecting procedure using linear response theory with forcing uncertainty

Here we follow the same procedure to project the ice loss of Antarctica in response to basal ice shelf melting as described in Levermann et al. (2014). In order to be able to compare to the previous results we use the same forcing data as in the 2014 publication. The only thing that changed is the ice sheet models that were used to compute the projections. The model initial states are those published in the initMIP intercomparison project for Antarctica (Seroussi et al., 2019). All other aspects of the projections, i.e. the procedure and the data that were used to force the models, are the same. We provide projections of the basal-melt-induced ice discharge from Antarctica for the four different carbon dioxide concentration scenarios (RCP2.6, RCP4.5, RCP6.0, RCP8.5; RCP is short for Representative Concentration Pathway; Moss et al., 2010). Here the RCP8.5 scenario represents a future evolution with increasing carbon emissions as seen in the past decades, while the RCP2.6 scenario represents one possible path that keeps the Paris Climate Agreement (United Nations, 2015) under certain conditions and thus keeps the global mean temperature increase below $2\,°C$ of global warming (Schleussner et al., 2016).

In a nutshell the method follows the schematic in Fig. 1: in order to provide a statistical estimate of the basal-melt-induced sea level contribution of Antarctica an ensemble of 20 000 basal melt forcing time series for the ice sheet models is created. Instead of forcing each model with each of the 20 000 forcing time series, the modelling groups carried out a specific simulation with a constant additional basal melt forcing of $8\,m\,yr^{-1}$, which was switched on at the beginning of the experiment after initialization and then kept constant for 200 years. The time derivative of the resulting sea level response of the specific ice sheet model within this experiment was used as a response function to a delta-distribution forcing of the ice sheet. Following the concept of a linear response theory this response function is convoluted with each of the forcing time series in order to estimate this ice sheet model's response to this specific forcing time series. Thereby it is possible to provide estimates of the response of 16 ice sheet models to 20 000 different basal melt forcing time series.

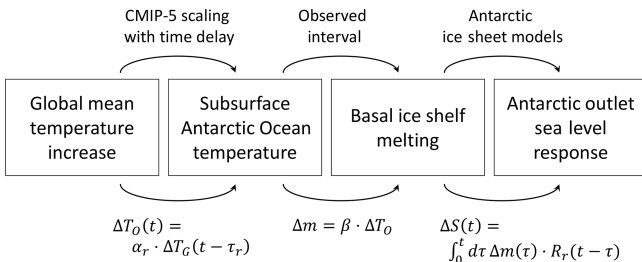

**Figure 1.** Schematic of the projection procedure: global mean temperature increase, $\Delta T_{G}$, is transformed into a subsurface warming around Antarctica, $\Delta T_{O}$, with a scaling coefficient, $\alpha_{r}$, and a time delay, $\tau_{r}$, both of which are derived for each of the five Antarctic outlet regions from 19 CMIP5 models. The basal ice shelf melting rate, $\Delta m$, is then derived by multiplying the subsurface oceanic temperature with a basal melt sensitivity $\beta$. This sensitivity is randomly chosen from the observed interval. The basal melt rate is then convoluted with the ice sheet response function of the specific region, $R_{r}$, to obtain the time series of this Antarctic outlet region.

The ensemble of basal melt forcing time series was created as follows (Fig. 1). Each ensemble member represents three random selections. First, a time series of the global mean temperature evolution from 1850 to 2100 is selected from an ensemble of 600 simulations of the MAGICC 6.0 emulator. These time series are all consistent with the observed warming path and the future carbon concentration pathway for which the sea level projection is computed (i.e. RCP2.6, 4.5, 6.0, or 8.5). Secondly, one of 19 CMIP5 climate models is selected in order to obtain a relation between the global mean surface warming and subsurface ocean warming which is forcing the Antarctic ice sheet. The subsurface ocean warming was computed in the five different regions around Antarctica shown in Fig. 2. In order to translate the global warming time series into a subsurface ocean warming time series for the different Antarctic regions a correlation coefficient in combination with a time delay was computed for each of the CMIP5 models. Thirdly, the subsurface ocean warming signal was multiplied with a value from the observed interval of melting sensitivities of the ice shelves. This way each surface warming signal is translated into a basal ice shelf melting signal in each of the Antarctic basins.

The random selection from 600 warming signals and 19 oceanic scaling functions is combined with a randomly uniform selection from the observed basal melt sensitivity interval to an ensemble of 20 000 time series for each emission scenario. The statistics of these time series are provided in Fig. 3.

Each ensemble member of these ice sheet forcing time series is then convoluted with the linear response function of the ice sheet model of the respective Antarctic region to obtain an estimate of the sea level contribution of this Antarctic sector to the global warming signal. This procedure is carried out for each member of the ensemble to obtain statistics of

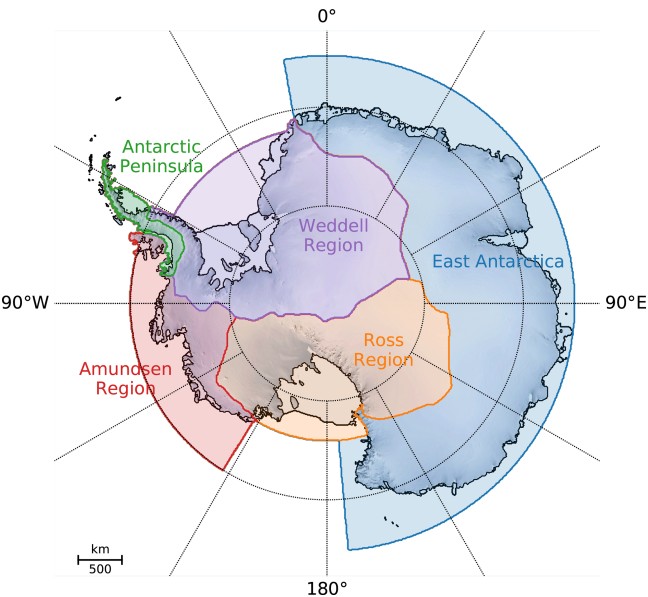

**Figure 2.** Oceanic regions in which the basal ice shelf melting was applied.

the sea level contribution of Antarctica from basal ice shelf melting.

In summary, for each emission scenario the procedure works as follows (each of the items is described in more detail below and in Levermann et al., 2014).

1. Randomly select a global mean temperature realization of the respective RCP scenario from the 600 MAGICC 6.0 realizations constrained by the observed temperature record. The time series start in 1850 and end in 2100.

2. Randomly select one of 19 CMIP5 models in order to obtain a scaling factor and a time delay for the relation between global mean surface air temperature and subsurface ocean warming in the respective regional sector in the Southern Ocean.

3. Randomly select a melting sensitivity in order to scale the regional subsurface warming outside the cavity of the Antarctic ice shelves onto basal ice shelf melting.

4. Select an ice sheet model that is forced via its linear response function with the time series of the forcing obtained from steps 1–3.

5. Compute the sea level contribution of this specific Antarctic ice sheet sector according to linear response theory.

6. Repeat steps 1–5 20 000 times with different random selections in each of the steps in order to obtain a probability distribution of the sea level contribution of each Antarctic sector and each carbon emission scenario.

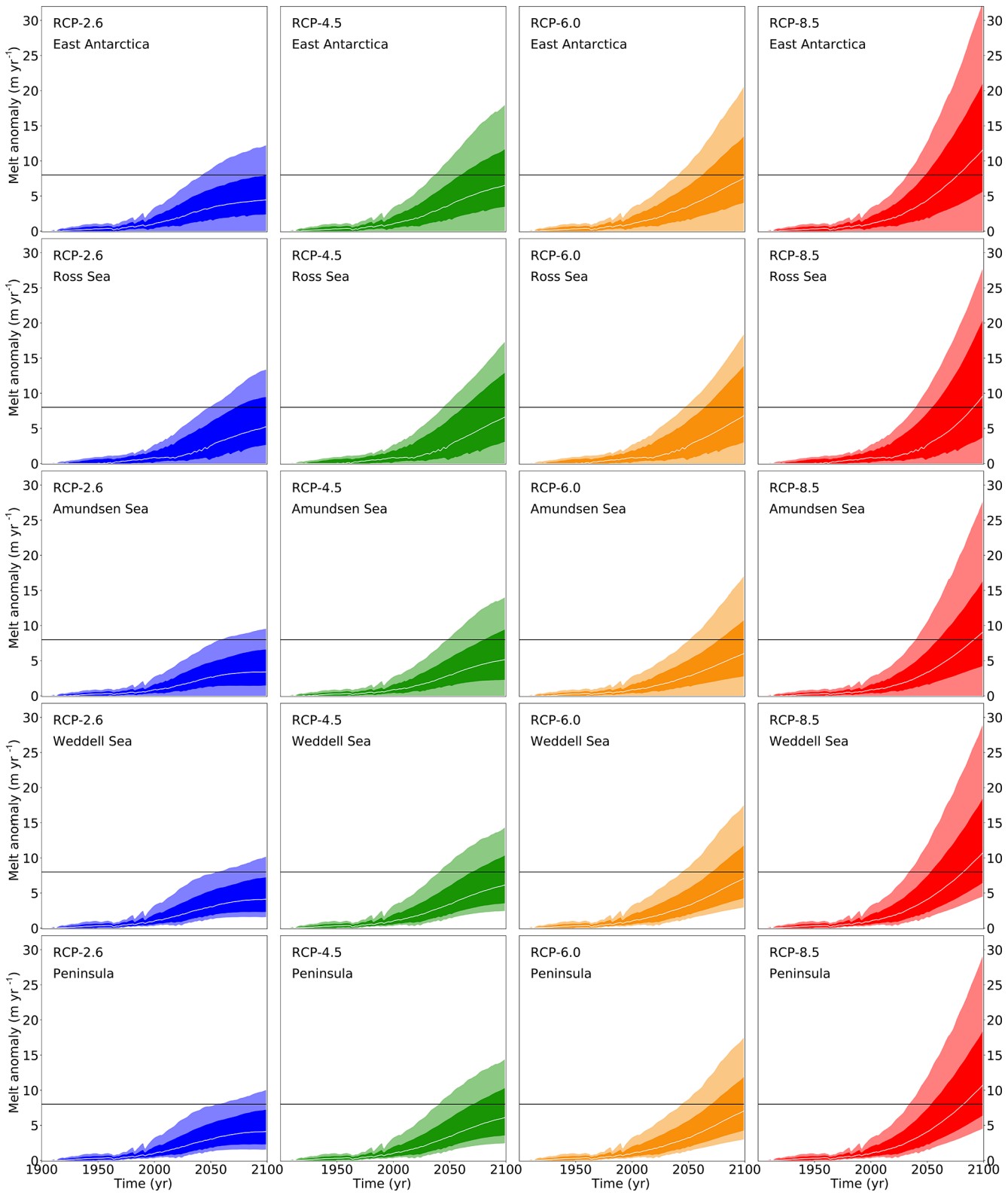

**Figure 3.** Projected basal melt rates following Sect. 2. The experiment used here for all the ice sheet models is the one with an additional $8\,\mathrm{m\,yr}^{-1}$ of basal melting (black horizontal line in each panel). It is the experiment that is closest to the projected basal melt rates, which fosters the applicability of the linear response theory.

Thus, the 20 000 selections are obtained by randomly choosing one temperature time series, one CMIP5 ocean model, one melt sensitivity, and one ice sheet model. The procedure is also used for each of the ice sheet models separately. In this case the random selection in step 4 is replaced by a fixed selection of the model. The procedure is illustrated in Fig. 1. For the computation of the total sea level contribution from all Antarctic sectors together, the forcing is selected consistently for all sectors. That means that for each of the 20 000 computations of the sea level contribution one global mean temperature realization is selected, as is one ocean model for the subsurface temperature scaling and one basal melt sensitivity. Although there are other possibilities, this approach was chosen because it preserves the forcing structure as provided by the ocean models. Details of steps 1–5 are given in the upcoming subsections.

## 2.1  Surface temperature scenario ensemble

We use the Representative Concentration Pathways (RCPs) (Meinshausen et al., 2011; Moss et al., 2010). The range of possible changes in global mean temperature that result from each RCP is obtained by constraining the response of the emulator model MAGICC 6.0 (Meinshausen et al., 2011) with the observed temperature record. This procedure has been used in several studies and aims to cover the possible global climate response to specific greenhouse gas emission pathways, including the carbon cycle feedbacks (e.g. Meinshausen et al., 2009). Here we use a set of 600 time series of global mean temperature from the year 1900 to 2100 for each RCP that cover the full range of future global temperature changes. Compare to Levermann et al. (2014) for details.

## 2.2  Subsurface oceanic temperature scaling

We use the simulations of the Coupled Model Intercomparison phase 5 (CMIP5) (Taylor et al., 2012) to obtain a scaling relationship between the anomalies of the global mean temperature and the anomalies of the oceanic subsurface temperature for each model. This has been carried out for the CMIP3 experiments (Winkelmann et al., 2012) and was repeated for the CMIP5 climate models in Levermann et al. (2014). The scaling approach is based on the assumption that anomalies of the ocean temperatures resulting from global warming scale with the respective anomalies in global mean temperature, with some time delay between the signals. We use oceanic temperatures from the subsurface at the mean depth of the ice shelf underside (Table 1) in each sector (Fig. 2) to capture the conditions at the entrance of the ice shelf cavities. As a small difference to the previous publication we modelled the Antarctic Peninsula separately with the ice sheet models. In order to be able to keep the same forcing we use, however, the same oceanic scaling as in the Amundsen region, which was the approach in the previous publication. The surface warming signal, $\Delta T_{\mathrm{G}}(t)$, needs to

**Table 1.** Mean depth of ice shelves in the different regions denoted in Fig. 2 as computed from Le Brocq et al. (2010), consistent with the previous study (Levermann et al., 2014) in order to make the results comparable. Oceanic temperature anomalies were averaged vertically over a range of 100 m around these depths.

| Region | Depth (m) |
|---|---|
| East Antarctica | 369 |
| Ross Sea | 312 |
| Amundsen Sea | 305 |
| Weddell Sea | 420 |
| Peninsula | 420 |

be transported to depth; therefore, the best linear regression is found with a time delay between the changes in global mean surface air temperature and subsurface oceanic temperatures, i.e.

$$\Delta T_{\mathrm{O}}(t) = \alpha_{\mathrm{r}} \cdot \Delta T_{\mathrm{G}}(t - \tau), \tag{1}$$

where $\tau$ is a CMIP5-model- and region-specific time delay.

For the probabilistic projections the scaling coefficients are randomly drawn from the 19 provided CMIP5 models. This approach does not account for changes due to abrupt ocean circulation changes (Hellmer et al., 2012), but the assumption is consistent with the linear response assumption underlying this study, and the correlation coefficients obtained for the 19 CMIP5 models used here are overall relatively high for each of the oceanic regions (Tables 2–5). In any case it is crucial to keep this limitation in mind when interpreting the results.

## 2.3  Sensitivity of basal ice shelf melting

In order to translate the ocean temperature changes into additional basal ice shelf melting for the five regions, we apply a basal melt sensitivity $\beta$ in a linear scaling approach, i.e.

$$\Delta m = \beta \cdot \Delta T_{\mathrm{O}}. \tag{2}$$

While great advances have been made in the past years bringing together observations and measurements of Southern Ocean properties (e.g. Schmidtko et al., 2014) as well as sub-shelf melt rates and volume loss from Antarctic ice shelves (e.g. Paolo et al., 2015; Rignot et al., 2013), the relation between oceanic warming and changes in basal melting is still subject to high uncertainties (Paolo et al., 2015).

Furthermore, some of the observed changes in sub-shelf melting are likely caused by changes in the ocean circulation rather than warming due to anthropogenic climate change (Hillenbrand et al., 2017; Jenkins et al., 2018). The recently observed ice loss in the Amundsen region, for instance, has been linked to the inflow of comparably warm circumpolar deep water into the ice shelf cavities (e.g. Hellmer et al., 2017; Pritchard et al., 2012). Similarly, the observed thinning in the Totten region in East Antarctica is largely driven

**Table 2.** East Antarctic sector: scaling coefficients, $\alpha_r$, and time delay, $\tau_r$, between increases in global mean temperature and subsurface ocean temperature anomalies.

| Model | Coeff. | $r^2$ | $\tau$ | Coeff. | $r^2$ |
|---|---|---|---|---|---|
| | without $\tau$ | | (yr) | with $\tau$ | |
| ACCESS1-0 | 0.20 | 0.92 | 30 | 0.35 | 0.94 |
| ACCESS1-3 | 0.27 | 0.92 | 0 | 0.27 | 0.92 |
| BNU-ESM | 0.35 | 0.92 | 0 | 0.35 | 0.92 |
| CanESM2 | 0.21 | 0.96 | 0 | 0.21 | 0.96 |
| CCSM4 | 0.13 | 0.96 | 5 | 0.13 | 0.97 |
| CESM1-BGC | 0.12 | 0.94 | 25 | 0.17 | 0.95 |
| CESM1-CAM5 | 0.15 | 0.94 | 0 | 0.15 | 0.94 |
| CSIRO-Mk3-6-0 | 0.22 | 0.93 | 15 | 0.28 | 0.94 |
| FGOALS-s2 | 0.17 | 0.90 | 55 | 0.41 | 0.94 |
| GFDL-CM3 | 0.21 | 0.89 | 35 | 0.39 | 0.93 |
| HadGEM2-ES | 0.23 | 0.95 | 0 | 0.23 | 0.95 |
| INMCM4 | 0.55 | 0.97 | 0 | 0.55 | 0.97 |
| IPSL-CM5A-MR | 0.14 | 0.89 | 0 | 0.14 | 0.89 |
| MIROC-ESM-CHEM | 0.11 | 0.89 | 0 | 0.11 | 0.89 |
| MIROC-ESM | 0.09 | 0.85 | 50 | 0.24 | 0.88 |
| MPI-ESM-LR | 0.20 | 0.94 | 15 | 0.26 | 0.95 |
| MRI-CGCM3 | 0.26 | 0.94 | 0 | 0.26 | 0.94 |
| NorESM1-M | 0.15 | 0.76 | 0 | 0.15 | 0.76 |
| NorESM1-ME | 0.15 | 0.74 | 60 | 0.49 | 0.85 |

**Table 4.** Amundsen Sea sector: scaling coefficients, $\alpha_r$, and time delay, $\tau_r$, between increases in global mean temperature and subsurface ocean temperature anomalies.

| Model | Coeff. | $r^2$ | $\tau$ | Coeff. | $r^2$ |
|---|---|---|---|---|---|
| | without $\tau$ | | (yr) | with $\tau$ | |
| ACCESS1-0 | 0.17 | 0.86 | 0 | 0.17 | 0.86 |
| ACCESS1-3 | 0.30 | 0.94 | 0 | 0.30 | 0.94 |
| BNU-ESM | 0.37 | 0.88 | 30 | 0.56 | 0.92 |
| CanESM2 | 0.15 | 0.83 | 30 | 0.24 | 0.88 |
| CCSM4 | 0.22 | 0.89 | 0 | 0.22 | 0.89 |
| CESM1-BGC | 0.19 | 0.92 | 0 | 0.19 | 0.92 |
| CESM1-CAM5 | 0.12 | 0.92 | 0 | 0.12 | 0.92 |
| CSIRO-Mk3-6-0 | 0.16 | 0.79 | 30 | 0.28 | 0.83 |
| FGOALS-s2 | 0.24 | 0.90 | 55 | 0.54 | 0.93 |
| GFDL-CM3 | 0.26 | 0.81 | 35 | 0.49 | 0.85 |
| HadGEM2-ES | 0.23 | 0.70 | 0 | 0.23 | 0.70 |
| INMCM4 | 0.67 | 0.90 | 0 | 0.67 | 0.90 |
| IPSL-CM5A-MR | 0.07 | 0.22 | 90 | 0.44 | 0.45 |
| MIROC-ESM-CHEM | 0.12 | 0.74 | 5 | 0.13 | 0.75 |
| MIROC-ESM | 0.11 | 0.55 | 60 | 0.35 | 0.61 |
| MPI-ESM-LR | 0.27 | 0.80 | 5 | 0.29 | 0.82 |
| MRI-CGCM3 | 0.00 | 0.02 | 85 | 0.00 | 0.04 |
| NorESM1-M | 0.30 | 0.94 | 0 | 0.30 | 0.94 |
| NorESM1-ME | 0.31 | 0.89 | 0 | 0.31 | 0.89 |

**Table 3.** Ross Sea sector: scaling coefficients, $\alpha_r$, and time delay, $\tau_r$, between increases in global mean temperature and subsurface ocean temperature anomalies.

| Model | Coeff. | $r^2$ | $\tau$ | Coeff. | $r^2$ |
|---|---|---|---|---|---|
| | without $\tau$ | | (yr) | with $\tau$ | |
| ACCESS1-0 | 0.17 | 0.86 | 0 | 0.17 | 0.86 |
| ACCESS1-3 | 0.30 | 0.94 | 0 | 0.30 | 0.94 |
| BNU-ESM | 0.37 | 0.88 | 30 | 0.56 | 0.92 |
| CanESM2 | 0.15 | 0.83 | 30 | 0.24 | 0.88 |
| CCSM4 | 0.22 | 0.89 | 0 | 0.22 | 0.89 |
| CESM1-BGC | 0.19 | 0.92 | 0 | 0.19 | 0.92 |
| CESM1-CAM5 | 0.12 | 0.92 | 0 | 0.12 | 0.92 |
| CSIRO-Mk3-6-0 | 0.16 | 0.79 | 30 | 0.28 | 0.83 |
| FGOALS-s2 | 0.24 | 0.90 | 55 | 0.54 | 0.93 |
| GFDL-CM3 | 0.26 | 0.81 | 35 | 0.49 | 0.85 |
| HadGEM2-ES | 0.23 | 0.70 | 0 | 0.23 | 0.70 |
| INMCM4 | 0.67 | 0.90 | 0 | 0.67 | 0.90 |
| IPSL-CM5A-MR | 0.07 | 0.22 | 90 | 0.44 | 0.45 |
| MIROC-ESM-CHEM | 0.12 | 0.74 | 5 | 0.13 | 0.75 |
| MIROC-ESM | 0.11 | 0.55 | 60 | 0.35 | 0.61 |
| MPI-ESM-LR | 0.27 | 0.80 | 5 | 0.29 | 0.82 |
| MRI-CGCM3 | 0.00 | 0.02 | 85 | −0.07 | 0.04 |
| NorESM1-M | 0.30 | 0.94 | 0 | 0.30 | 0.94 |
| NorESM1-ME | 0.31 | 0.89 | 0 | 0.31 | 0.89 |

**Table 5.** Weddell Sea sector and Antarctic Peninsula: scaling coefficients, $\alpha_r$, and time delay, $\tau_r$, between increases in global mean temperature and subsurface ocean temperature anomalies.

| Model | Coeff. | $r^2$ | $\tau$ | Coeff. | $r^2$ |
|---|---|---|---|---|---|
| | without $\tau$ | | (yr) | with $\tau$ | |
| ACCESS1-0 | 0.18 | 0.77 | 20 | 0.26 | 0.79 |
| ACCESS1-3 | 0.09 | 0.76 | 15 | 0.12 | 0.77 |
| BNU-ESM | 0.28 | 0.83 | 20 | 0.36 | 0.84 |
| CanESM2 | 0.14 | 0.74 | 45 | 0.32 | 0.80 |
| CCSM4 | 0.14 | 0.91 | 5 | 0.15 | 0.92 |
| CESM1-BGC | 0.14 | 0.90 | 0 | 0.14 | 0.90 |
| CESM1-CAM5 | 0.16 | 0.85 | 0 | 0.16 | 0.85 |
| CSIRO-Mk3-6-0 | 0.00 | 0.28 | 0 | 0.00 | 0.28 |
| FGOALS-s2 | 0.18 | 0.89 | 60 | 0.45 | 0.93 |
| GFDL-CM3 | 0.23 | 0.85 | 25 | 0.37 | 0.89 |
| HadGEM2-ES | 0.25 | 0.62 | 0 | 0.25 | 0.62 |
| INMCM4 | 0.59 | 0.83 | 0 | 0.59 | 0.83 |
| IPSL-CM5A-MR | 0.02 | 0.04 | 95 | 0.14 | 0.12 |
| MIROC-ESM-CHEM | 0.23 | 0.85 | 0 | 0.23 | 0.85 |
| MIROC-ESM | 0.23 | 0.78 | 0 | 0.23 | 0.78 |
| MPI-ESM-LR | 0.16 | 0.70 | 40 | 0.31 | 0.73 |
| MRI-CGCM3 | 0.08 | 0.04 | 0 | 0.08 | 0.04 |
| NorESM1-M | 0.12 | 0.79 | 0 | 0.12 | 0.79 |
| NorESM1-ME | 0.12 | 0.68 | 20 | 0.16 | 0.73 |

by changes in the surrounding ocean circulation (Greenbaum et al., 2015; Wouters et al., 2015).

In our simplified approach, we therefore draw the melt sensitivity parameter with equal probability from an empirically based interval between 7 and $16 \, \mathrm{m \, a^{-1} \, K^{-1}}$ (based on Jenkins, 1991; Payne et al., 2007). While this approach neglects the complex patterns arising for observed basal melt rates in Antarctica, it is consistent with the response function methodology adopted here. Note that we are applying melt rate anomalies to derive the response functions – the ice sheet model simulations still display a wide range of total melt rates over space, with generally higher melting near the grounding line and lower melting or even refreezing towards the ice shelf front. This is consistent with the vertical overturning circulation typically found in ice shelf cavities (Lazeroms et al., 2018; Olbers and Hellmer, 2010; Reese et al., 2018b).

Combining the global mean temperature time series of Sect. 2.1 and the CMIP5 oceanic scaling of Sect. 2.2 with the basal melt sensitivity described here in a probabilistic way, i.e. by choosing an ensemble of 20 000 combinations of each of these three components, yields the basal melt time series in Fig. 3. The horizontal black line depicts the $8 \, \mathrm{m \, yr^{-1}}$ level. The basal melt time series are scattered around this level. For the projections we will thus use the switch-on experiments with $8 \, \mathrm{m \, yr^{-1}}$ of additional basal melt as described below. This is the most balanced choice to span the range of simulations, with $4 \, \mathrm{m \, yr^{-1}}$ being too low for most of the RCP8.5 scenario and $16 \, \mathrm{m \, yr^{-1}}$ being too high for the majority of scenarios and ensemble samples.

The reason for carrying out experiments in which a constant additional basal melt forcing is applied for a period of 200 years is that this allows us to easily derive linear response functions for the different ice sheet models in the different regions as described in the next subsection.

## 2.4 Deriving the ice sheet response function

The core of the projections of the future sea level contribution from Antarctic basal ice shelf melting is composed of simulations with 16 ice sheet models. The models were forced with a constant additional basal ice shelf melting of $8 \, \mathrm{m \, yr^{-1}}$. The forcing was applied homogeneously in each of the five oceanic sectors separately (Fig. 2). The regions were chosen to avoid ice dynamic interference between the regions on the timescale of this century. In order to check this, additional simulations with all regions forced simultaneously were carried out by some of the modelling groups (all data are provided as a Supplement to this paper). These simulations showed that any possible non-linear interactions between the flow of the different basins which do exist on longer timescales (Martin et al., 2019) are negligible on the timescale of 200 years used here and will not be considered any further in this study. For comparison additional simulations with 4 and $16 \, \mathrm{m \, yr^{-1}}$ were carried out. This is dis-

cussed below. A number of modelling groups carried out further simulations with 1, 2, and $32 \, \mathrm{m \, yr^{-1}}$ basal melt rates. Although these simulations are highly interesting, a full discussion of their results is beyond the scope of this publication. The results of the $32 \, \mathrm{m \, yr^{-1}}$ simulations are provided in Figs. S1–S4 in the Supplement. Here we aim at providing an estimate of the future sea level contribution from Antarctic ice discharge and the uncertainty that is associated with the external forcing.

One of the strongest assumptions of the projections computed here is that of a linear response of the ice sheet dynamics to external forcing. This, however, does not mean that it is assumed that the ice discharge is increasing linearly with time. It merely assumes that increasing the magnitude of the forcing by a specific factor will increase the magnitude of the response of the ice sheet by the same factor. The temporal evolution of the ice sheet is given by a temporarily varying response function. The response function, $R(t)$, is defined as the response of the system to a delta-peak forcing. It could be estimated by measuring the response of the ice sheet to a 1-year basal melt forcing of $1 \, \mathrm{m \, yr^{-1}}$, which would correspond to a unit forcing for a short period of time. Once the response function is known the assumption is that the response to any given forcing, $m(t)$, can be obtained by linear superposition, which in a time-continuous situation translates into a convolution of the response function with the forcing:

$$S(t) = \int_0^t \mathrm{d}\tau \, m(\tau) \cdot R(t - \tau), \tag{3}$$

where $S(t)$ is the sea level contribution from ice discharge and, $t$ is time starting from a period prior to the beginning of a significant forcing. From Eq. (3) it is clear that the response function can also be obtained from a Heaviside forcing whereby basal melt is switched on to a constant value, $\mu$, at a specific time and then kept constant as was done here. In that case the observed response, $A_\mu(t)$, is simply the time integral of the response function:

$$A_\mu(t) = \mu \cdot \int_0^t \mathrm{d}\tau \, R(\tau). \tag{4}$$

The response functions for each of the ice sheet models use the fixed Heaviside forcing $\mu = 8 \, \mathrm{m \, yr^{-1}}$ and then take the time derivative of the response $A_{8 \, \mathrm{m \, yr^{-1}}}(t)$ and divide by $8 \, \mathrm{m \, yr^{-1}}$. Due to the relatively strong inertia of ice sheet models this approach generally yields more robust results compared to a delta-peak approach, which is why we have followed this path here. Another option which is often used in solid-state physics to obtain the response functions (for example, their oscillatory excitations) is by forcing the system with white noise. Fourier transformation of Eq. (3) will then transform the convolution into a simple product, and the white noise becomes a constant in Fourier space. The Fourier

transform of the response divided by this constant is then simply the Fourier transform of the response function. This approach, however, is not helpful to obtain a short-term response to a slow-moving system such as an ice sheet.

## 2.5 Description of the ice sheet models

The ice sheet models used here all take part in the initMIP intercomparison project for Antarctica (Seroussi et al., 2019) within the overall ISMIP6 initiative (Goelzer et al., 2018; Nowicki et al., 2016). Since the description of their respective ability to reproduce the present ice dynamics of Antarctica is a study in its own, we refer to the corresponding model description papers and provide only a brief description of each of the model in Appendix A.

## 2.6 Validity of the linearity assumption

In order to assess the validity of the linearity assumption, we plotted in Fig. 4a–e the original simulations of each model for an $8 \, \mathrm{m \, yr^{-1}}$ additional basal melt forcing (black curves) which is held constant over 200 years. In addition, we plot the outcome of the $4 \, \mathrm{m \, yr^{-1}}$ experiment (blue solid curves) and the $16 \, \mathrm{m \, yr^{-1}}$ experiments (red solid curves) together with the $8 \, \mathrm{m \, yr^{-1}}$ experiments divided by 2 (blue dashed) and multiplied by 2 (red dashed). Generally the agreement is reasonable. The fact that the validity of the linearity assumption can be extended all the way to a doubling and halving of the forcing is extraordinary where it is true.

As a quasi-quantitative measure for the validity of the linearity assumption we computed an exponent $\alpha$ such that the curves

$$A_{4,\alpha}(t) \equiv \left(\frac{4 \, \mathrm{m \, yr^{-1}}}{8 \, \mathrm{m \, yr^{-1}}}\right)^{1+\alpha} \cdot A_8(t) = 2^{-(1+\alpha)} \cdot A_8(t) \qquad (5)$$

$$A_{16,\alpha}(t) \equiv \left(\frac{16 \, \mathrm{m \, yr^{-1}}}{8 \, \mathrm{m \, yr^{-1}}}\right)^{1+\alpha} \cdot A_8(t) = 2^{(1+\alpha)} \cdot A_8(t) \qquad (6)$$

have the least square error to their respective target functions $A_4(t)$ and $A_{16}(t)$. The values for $\alpha$ are provided for each model in each sector in Fig. 4a–e together with the respective curves as dotted lines. In the case of perfect linearity $\alpha = 0$. If $\alpha < 0$ a doubling of $A_8(t)$ yields a curve that is higher than $A_{16}(t)$; i.e. the model responds sub-linearly to basal melting. This also means that a halving of $A_8(t)$ is an overestimation of $A_4(t)$. This was the case for most models. As can be seen from the comparison of the curves with $\alpha$ correction and the original simulations, linearity can be assumed for the relatively short response time of 200 years and the forcing range applied in this study.

The term "no scaling" was used when no $-1 < \alpha < 2$ represented a valid minimum of the error; i.e. the different experiments are not linearly related. This is only the case for very small and noisy responses in the Antarctic Peninsula. The term "no data" means that the modelling group did not provide the corresponding data. For the computation of the sea level projections the $8 \, \mathrm{m \, yr^{-1}}$ experiments were used throughout this study.

The response function for each model and each region is given in Fig. 5a–e together with their 10-year running mean. The response function is unitless because it is a sea level rise $(\mathrm{m \, yr^{-1}})$ divided by basal melt rate $(\mathrm{m \, yr^{-1}})$. Note that this is the response the model would show for a short and sudden forcing of 1 year of $1 \, \mathrm{m \, yr^{-1}}$ additional basal ice shelf melting in the region. While some models show an instantaneous ice loss response (e.g. in East Antarctica the models AISM VUB, ISSM UCI, PISM VUW, and ÚA UNN), most models exhibit a more gradual increase in the ice loss over time. The temporal structure of the response is a result of the complexity of the ice dynamics and its interaction with the initial condition and the bed topography.

As can be seen from the basal melt projections in Fig. 3 the applied melt rates vary strongly around $8 \, \mathrm{m \, yr^{-1}}$. In the Supplement (Fig. S1a–e) the results for the $32 \, \mathrm{m \, yr^{-1}}$ switch-on experiments are provided for context for the models that have performed these experiments. The linearity assumption is not necessarily a good assumption in all cases, but in most cases the assumption is a reasonable approximation of how basal melting is responding to external forcing.

For the adaptive grid model BISI LBL, the scaling is shown for simulations with the finest horizontal resolution of 1000 m, while the projections are carried out with a simulation with the finest horizontal resolution of 500 m (shown as the black dashed curve in the BISI LBL panels in Fig. 4a–e). Due to computational constraints the linearity check had to be done at the slightly lower resolution (1000 m). As can be seen in the Fig. 4a–e there are some quantitative deviations between the higher- and lower-resolution simulations, but the results are not qualitatively different.

## 3 Hindcasting the observational record

The projections of sea level contributions from Antarctica due to basal melting underneath the ice shelves following the linear response theory were started in the year 1900 in order to make sure that no significant global mean temperature increase influences the outcome. Following the procedure described above and thereby using the combined equations of Fig. 1, the sea level contribution is computed from

$$S_{\mathrm{r}}(t) = \int_0^t \mathrm{d}\tau \beta \cdot \alpha_{\mathrm{r}} \cdot \Delta T_{\mathrm{G}}(\tau - \tau_{\mathrm{r}}) \cdot R(t - \tau)$$

$$= \alpha_{\mathrm{r}} \beta \cdot \int_f^t 0^t \mathrm{d}\tau \, \Delta T_{\mathrm{G}}(\tau - \tau_{\mathrm{r}}) \cdot R(t - \tau), \qquad (7)$$

with constants $\alpha_{\mathrm{r}}$, $\beta$, and $\tau_{\mathrm{r}}$ derived from observations or CMIP5 model results and the index, $R$, indicating the specific Antarctic forcing region (Fig. 2). We can then hind-

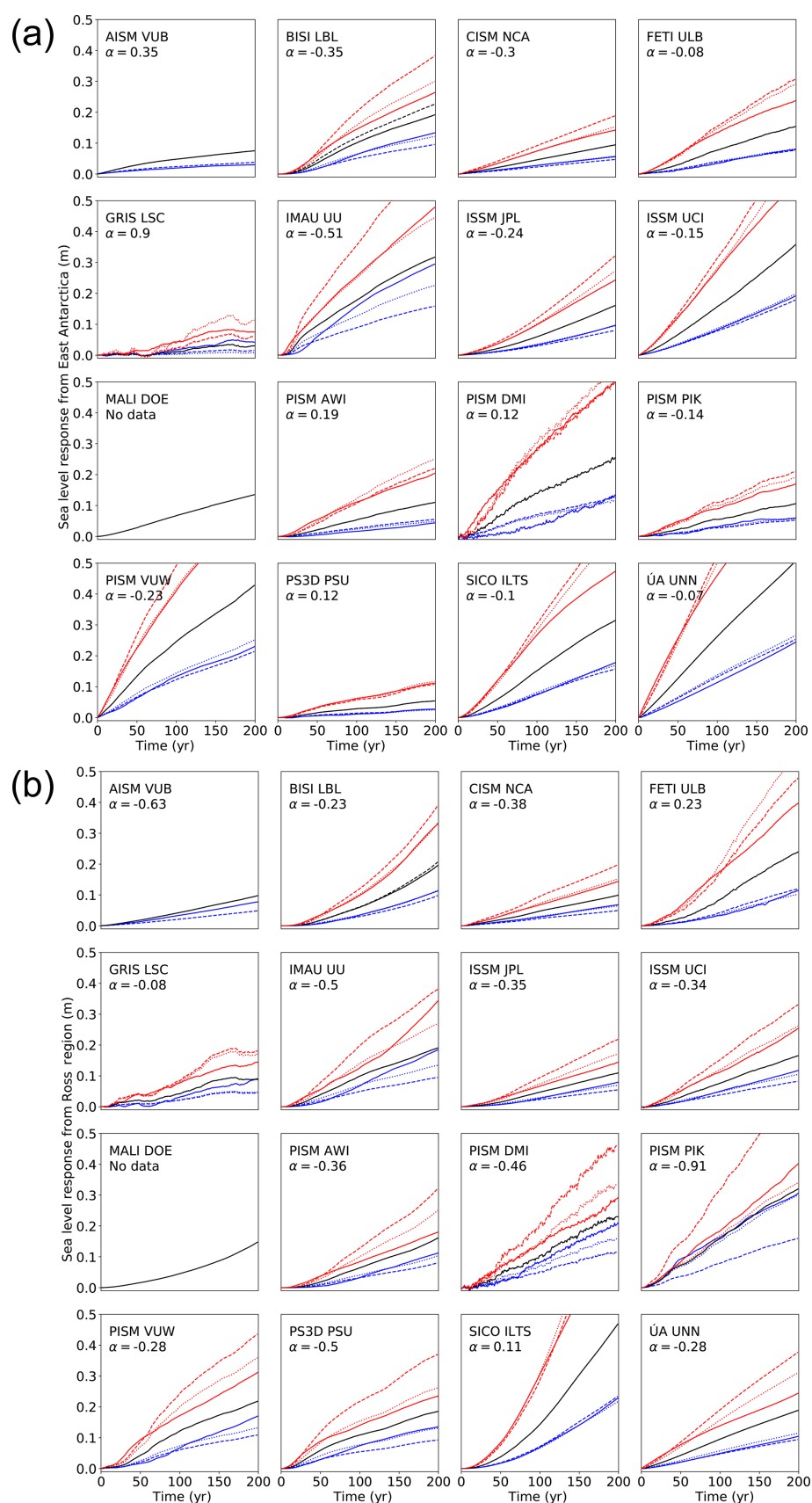

**Figure 4.**

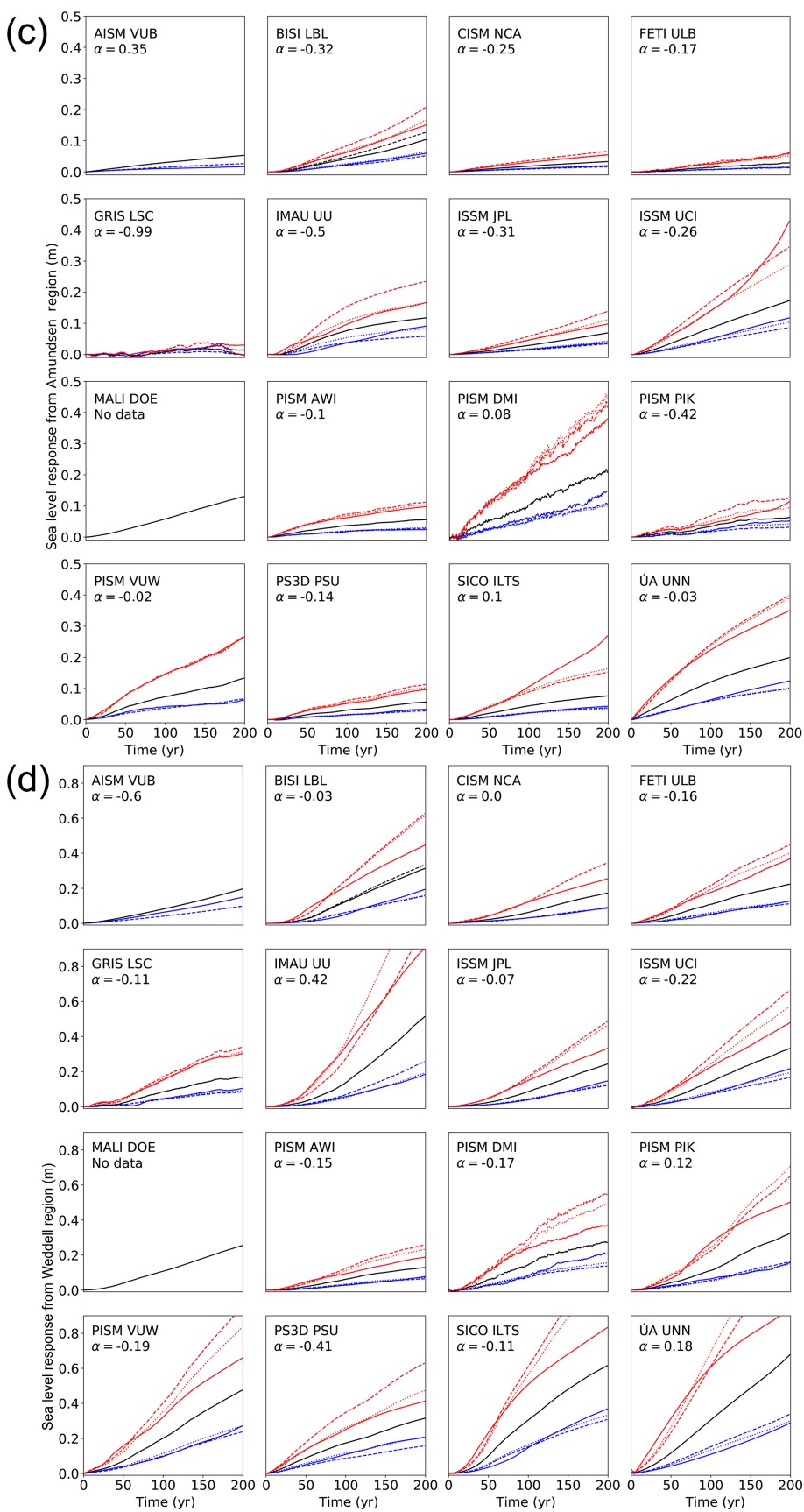

**Figure 4.**

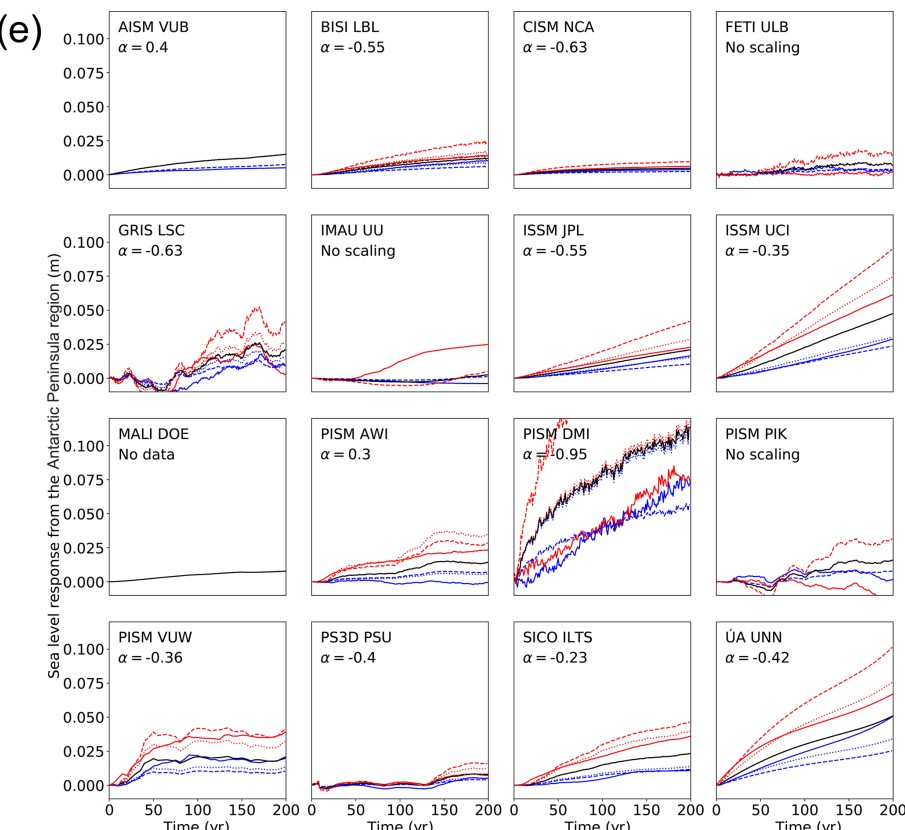

**Figure 4. (a)** Linearity check for East Antarctica. Response of ice sheet models to additional basal melting of $8 \, \text{m} \, \text{yr}^{-1}$ (solid black line) underneath all ice shelves in East Antarctica compared to $4 \, \text{m} \, \text{yr}^{-1}$ (solid blue line) and $16 \, \text{m} \, \text{yr}^{-1}$ (solid red line). In order to check the linearity of the response to the warming amplitude the dashed red line gives the times series of the response to $8 \, \text{m} \, \text{yr}^{-1}$ of basal melt multiplied by 2 and the dashed blue line the same but divided by 2. The dotted lines give the scaled response with the scaling exponent $\alpha$ (see Eqs. 5 and 6). A positive scaling exponent means that the ice sheet model responds super-linearly to basal ice shelf melting in this region. A negative $\alpha$ indicates a sub-linear response in this region. AISM VUB did not provide a $16 \, \text{m} \, \text{yr}^{-1}$ simulation. The black dashed line for BISI LBL represents the simulation with 500 m horizontal resolution that is used for the projections. The linearity is tested with a set of simulations at 1 km horizontal resolution. **(b)** Linearity check for the Ross region as in panel **(a)**. **(c)** Linearity check for the Amundsen region as in panel **(a)**. **(d)** Linearity check for the Weddell region as in panel **(a)**. **(e)** Linearity check for the Antarctic Peninsula as in panel **(a)**.

cast the observed sea level contribution between 1992 and 2017 and compare it to observations (Fig. 6). To this end we use the results by Shepherd et al. (2018), which do not differ significantly from earlier estimates (Shepherd et al., 2012). The time series of the median observed sea level contribution is given as a white line in Figs. 6 and 7 with the uncertainty range given in grey shading. The individual model results are given as the median and the likely range around this median (66th percentile around the median) as the full and dotted black lines. While individual models may deviate strongly from the observed range, the combination of all models shows a similar contribution for the time period 1992–2017 as was observed with a bias towards slightly higher ice loss (Figs. 6 and 7, Table 6).

An important issue regarding the comparison with observations (Fig. 6) is whether the individual models or individual projections should be weighted according to their ability to hindcast the observed contribution to global sea level rise.

One way to do this would be to compute the weight, $w_i$, of a specific computed time series (using a specific atmospheric temperature time series, a specific ocean model, and a specific melting sensitivity) as follows:

$$w_i = \frac{1}{N} \cdot e^{(\Delta S_i - \Delta S_{\text{obs}})^2 / 2\sigma}$$

where $\Delta S_{\text{obs}}$ is the observed median sea level contribution of Antarctica between 1992 and 2017 according to Shepherd et al. (2018), and $\sigma$ is the uncertainty of this estimate according to the same publication. The normalization factor $N$ would depend on the sample of computations compared. It would be chosen such that the sum over all realizations within a set is 1. Thus, the weight for a specific realization could be different if the contribution is computed for only one specific ice model or if it is computed for all ice models. We have decided against this kind of weighting for the simple reason that the comparison of a model–forcing combination to

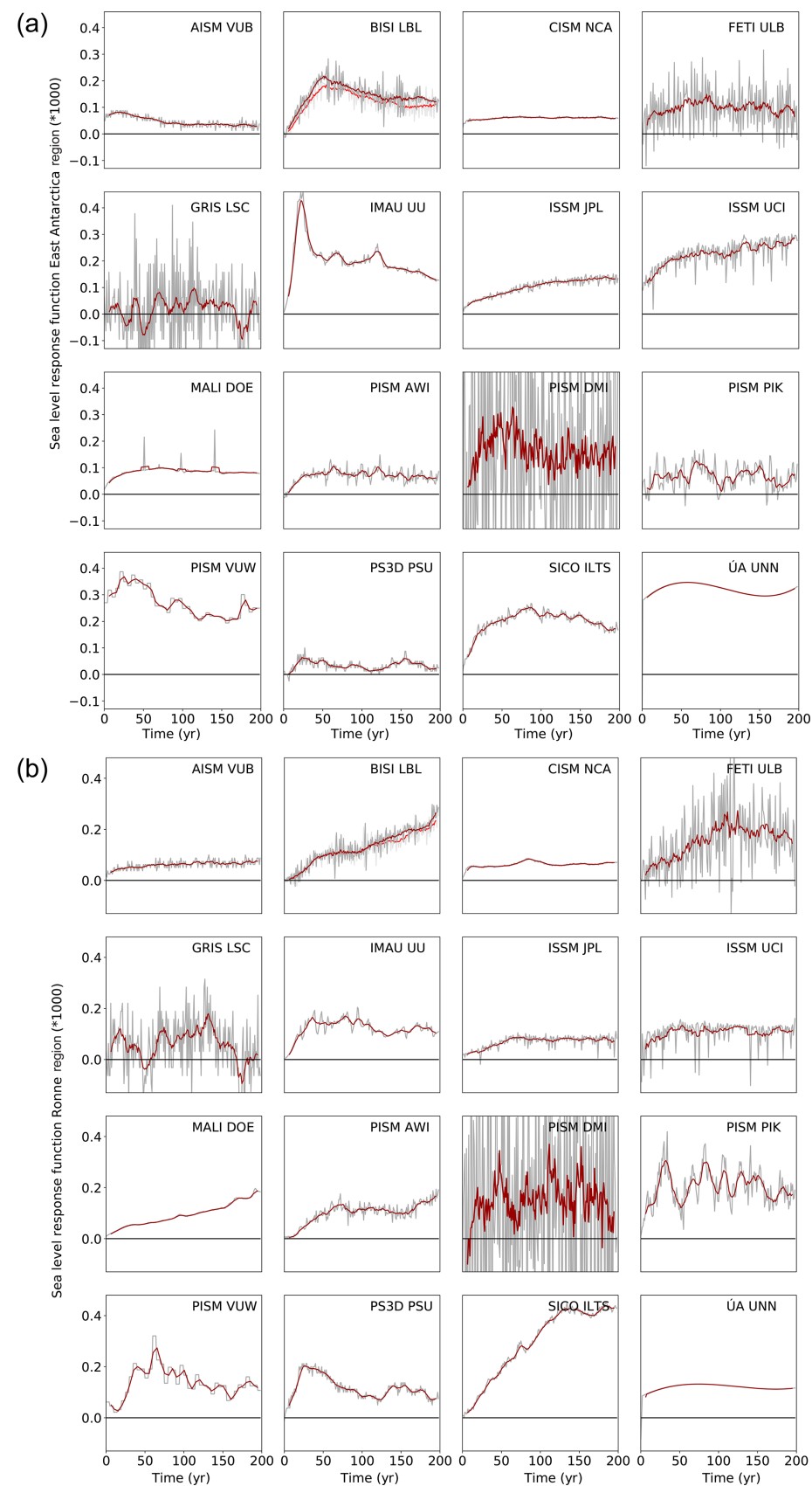

**Figure 5.**

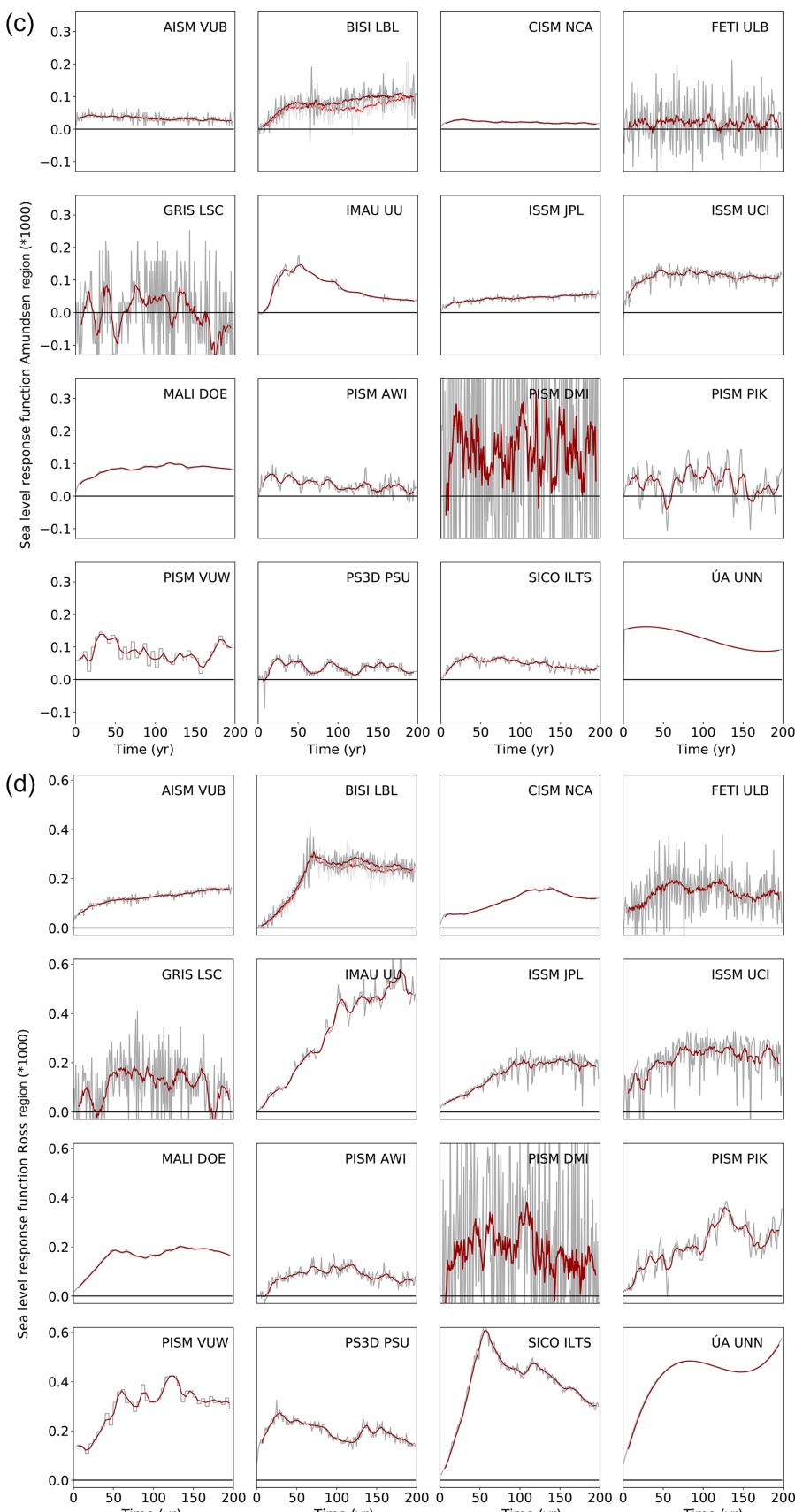

**Figure 5.**

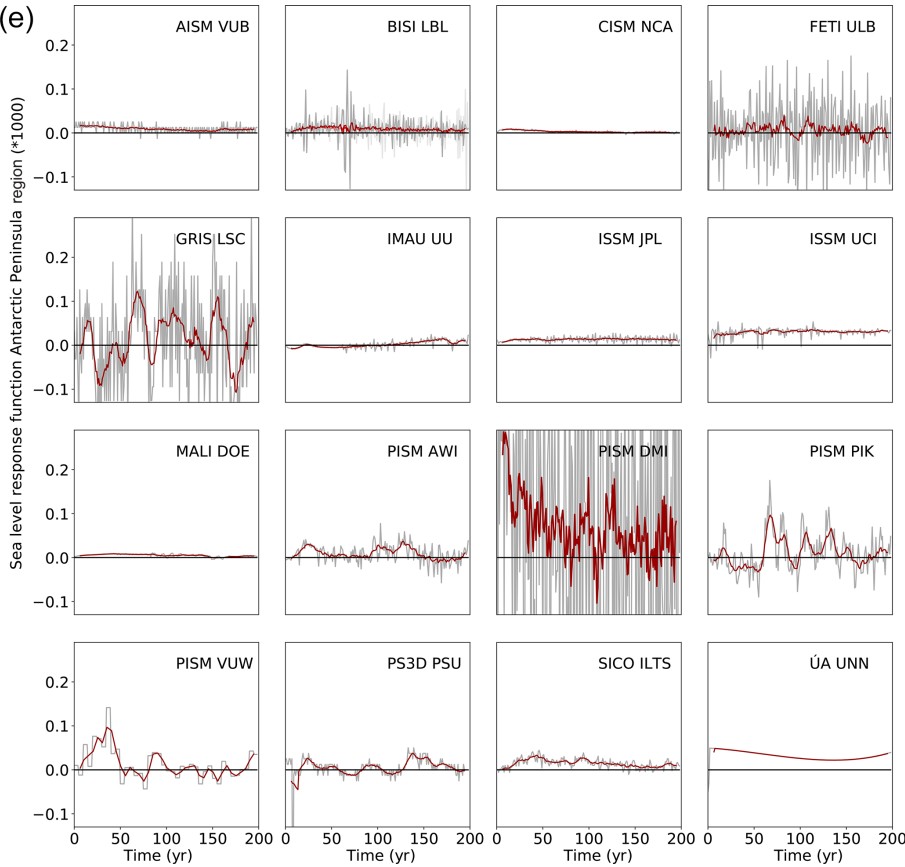

**Figure 5. (a)** Response function for the East Antarctica region. Response function computed from the time derivative of the response of the ice sheet models within the experiment with additional basal melting of $8\,\mathrm{m\,yr^{-1}}$ divided by $8\,\mathrm{m\,yr^{-1}}$. The response function is thus unitless in this specific case. The red line provides a 10-year running mean. For the BISI LBL `CEI` model the simulation with 1 km horizontal resolution (compared to the main simulation with 500 m horizontal resolution that is used for the projections) is also shown. These are the light grey lines and the light red 10-year running mean. **(b)** Response function for the Weddell Sea region as in panel **(a)**. **(c)** Response function for the Amundsen Sea region as in panel **(a)**. **(d)** Response function for the Ross Sea region as in panel **(a)**. **(e)** Response function for the Antarctic Peninsula region as in panel **(a)**.

reproduce the past does not reflect its ability to project the future. The reason for this is that the main contribution from Antarctica to the sea level rise since 1992 arose from a specific oceanic warming in the Amundsen Sea sector, which cannot be easily linked to the global mean temperature increase. It is definitely not reflected in the procedure that we apply here to obtain the forcing underneath the ice shelves (Fig. 1). Applying such a weighting would thus distort the results in an unjustified way.

The comparison is done here in order to illustrate the order of magnitude of the signal that is obtained by this procedure. Compared to earlier ice sheet models the newer generation is able to exhibit a dynamic behaviour that is at least of the same order of magnitude compared to observations. Here only positive temperature anomalies above the reference level are accounted for. That is because it cannot be claimed that a linear response as described in Eq. (7) can also capture a negative response, which would be due to processes

like refreezing. This may lead to a small positive bias in the initial period at the beginning of the 20th century and thereby to a small overestimation of the observed sea level contribution. Furthermore, the observations will include changes in surface mass balance, in particular an increase in snowfall, which is not captured by our approach. Thus, even though the comparison with observations seems to be compelling, it is not as strong a test as it might seem.

## 4 Projecting the 21st century sea level contribution of Antarctica from basal ice shelf melting

Finally, we compute the projections of Antarctica's contribution to future sea level rise using Eq. (3) following the schematic of Fig. 1 as described in Sect. 2. The overall Antarctic projections, including all uncertainty in basal melt forcing for each of the ice sheet models under the atmospheric $CO_2$ concentration path RCP2.6 and 8.5, are given

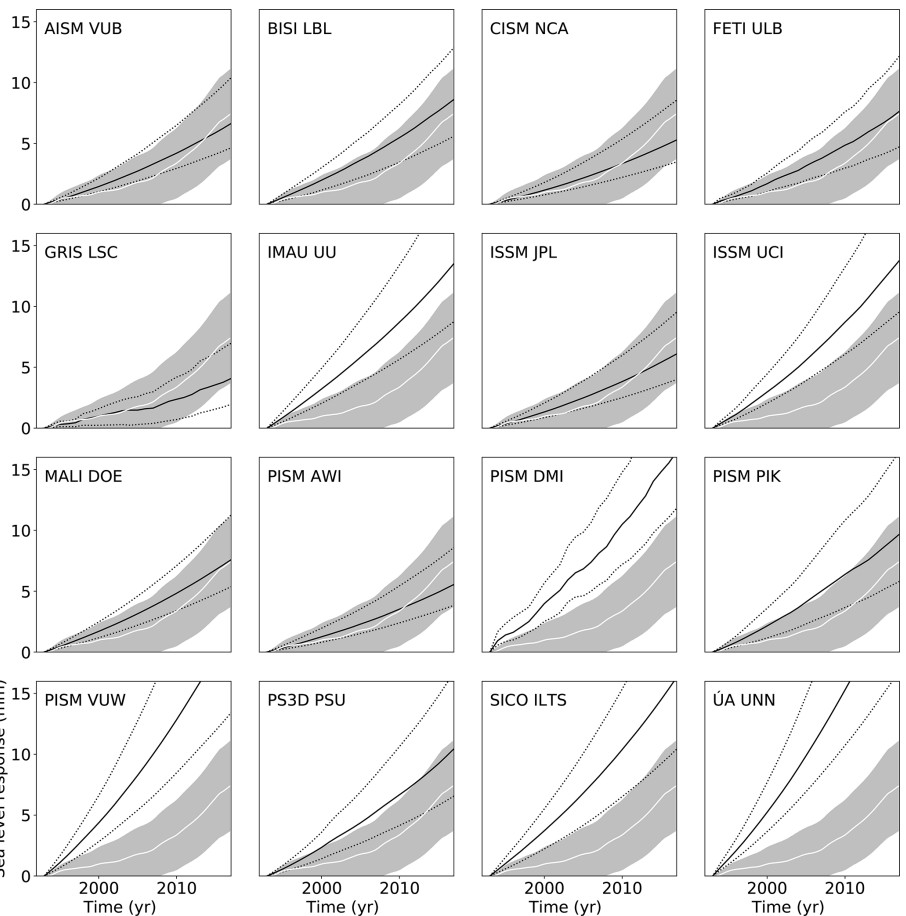

**Figure 6.** Hindcasting observed sea level contributions and modelled sea level contribution of Antarctica from the different ice sheet models. The solid black line represents the median contribution between 1992 and 2017 with the 66th percentile (first standard deviation) around the median. The grey shading represents the uncertainty range of the observed contribution of Antarctica (white solid line) following Shepherd et al. (2018).

in Figs. 8 and 9, respectively. The values for the median, the likely range (percentiles 16.6 and 83.3), and the very likely range (5th and 95th percentiles) are provided in Tables 7–10 for all four RCP scenarios for the year 2100.

5   The results for RCP8.5 for each of the five Antarctic regions are provided in Fig. 10a–e. The results differ between the different models. Overall, median contributions of around 5 cm come from the Weddell Sea sector and the East Antarctic, while the Ross and Amundsen Sea sectors have a median 10 contribution of around 2 cm, and the peninsula has the lowest median contribution. Although the largest median contributions arise from the Weddell Sea sector, the largest 95th percentile is found in the East Antarctic sector (Fig. 11). This is because the forcing onto the ice sheet is transported not with 15 a particular oceanic current but is mainly mixed to the ice shelves due to the overall coarse resolution of the CMIP5 climate models. It thus arrives everywhere, and the East Antarctic Ice Sheet has the most ice catchment area that is in direct contact with the ocean due to its size. In East Antarc- 20 tica four of the models have a stronger contribution than the

others (PISM VUW, ÚA UNN, IMAU UU, and ISSM UCI). For the three West Antarctic outlet regions the model results are more similar than for East Antarctica. Overall the models show quite similar responses to the forcing, and overall the uncertainty in the sea level response is dominated by the 25 uncertainty in the forcing.

There are a number of different reasons for relatively weak responses of some models in some regions. These reasons are as diverse as the models. It is beyond the scope of this article to provide a detailed analysis of the causes for the model 30 response differences. Here we discuss possible reasons for the deviations of the different models from the median in order to give some information that is specific to the different models and to provide some guidance on reading the results.

For the spin-up of the GRIS LSC model, an inversion pro- 35 cedure (Le clec'h et al., 2019b) was used to infer a map of the basal drag coefficient that minimizes the ice thickness mismatch with respect to the observations and the drift over a 200-year equilibrium simulation. The procedure is iterative and computationally cheap. Some errors compared to 40

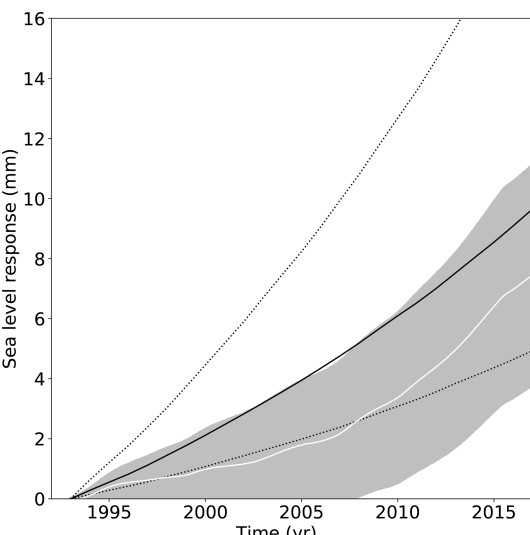

**Figure 7.** Hindcasting of all models combined with observed sea level contributions and modelled sea level contribution of Antarctica from the different ice sheet models. The solid black line represents the median contribution between 1992 and 2017 with the 66th percentile (first standard deviation) around the median. The grey shading represents the uncertainty range of the observed contribution of Antarctica (white solid line) following Shepherd et al. (2018).

**Table 7.** Percentiles of the probability distribution of the sea level contribution of Antarctica for different ice sheet models under the RCP2.6 climate scenario. The 50th percentile corresponds to the median; 16.6 %–83.3 % is the so-called "likely range" as denoted in the IPCC reports. The "very likely range" is given by the 5th–95th percentiles.

| RCP2.6 | Antarctica sea level contribution percentiles (m) | | | | |
|---|---|---|---|---|---|
| Model | 5 % | 16.6 % | 50 % | 83.3 % | 95 % |
| AISM VUB | 0.04 | 0.06 | 0.06 | 0.12 | 0.20 |
| BISI LBL | 0.06 | 0.09 | 0.14 | 0.20 | 0.34 |
| CISM NCA | 0.03 | 0.05 | 0.07 | 0.10 | 0.17 |
| FETI ULB | 0.05 | 0.07 | 0.11 | 0.16 | 0.26 |
| GRIS LSC | 0.02 | 0.04 | 0.06 | 0.08 | 0.13 |
| IMAU UU | 0.09 | 0.14 | 0.20 | 0.28 | 0.49 |
| ISSM JPL | 0.04 | 0.06 | 0.09 | 0.13 | 0.22 |
| ISSM UCI | 0.09 | 0.14 | 0.19 | 0.27 | 0.45 |
| MALI DOE | 0.06 | 0.08 | 0.11 | 0.15 | 0.26 |
| PISM AWI | 0.04 | 0.06 | 0.08 | 0.12 | 0.20 |
| PISM DMI | 0.11 | 0.16 | 0.22 | 0.31 | 0.52 |
| PISM PIK | 0.06 | 0.09 | 0.14 | 0.22 | 0.31 |
| PISM VUW | 0.13 | 0.18 | 0.26 | 0.38 | 0.62 |
| PS3D PSU | 0.06 | 0.09 | 0.14 | 0.20 | 0.30 |
| SICO ILTS | 0.12 | 0.16 | 0.25 | 0.35 | 0.60 |
| ÚA UNN | 0.16 | 0.22 | 0.32 | 0.46 | 0.76 |
| All models | 0.05 | 0.07 | 0.14 | 0.27 | 0.40 |

**Table 6.** Likely hindcast range of historical sea level contribution compared to the observed range.

| Observed and modelled contribution 1992 to 2017 | Antarctica sea level contribution percentiles (mm) | | |
|---|---|---|---|
| | 16.6 % | 50 % | 83.3 % |
| Observations | 3.7 | 7.4 | 11.1 |
| All models | 5.2 | 10.2 | 21.3 |
| AISM VUB | 4.9 | 7.0 | 11.0 |
| BISI LBL | 5.9 | 9.1 | 13.6 |
| CISM NCA | 3.7 | 5.6 | 9.0 |
| FETI ULB | 4.9 | 8.0 | 12.8 |
| GRIS LSC | 2.2 | 4.5 | 7.6 |
| IMAU UU | 9.2 | 14.3 | 22.5 |
| ISSM JPL | 4.2 | 6.5 | 10.1 |
| ISSM UCI | 10.2 | 14.7 | 22.5 |
| MALI DOE | 5.7 | 8.0 | 11.9 |
| PISM AWI | 4.1 | 5.9 | 9.1 |
| PISM DMI | 12.6 | 17.4 | 25.3 |
| PISM PIK | 6.1 | 10.3 | 17.5 |
| PISM VUW | 14.2 | 21.5 | 33.9 |
| PS3D PSU | 6.9 | 11.0 | 17.7 |
| SICO ILTS | 11.1 | 17.3 | 25.7 |
| ÚA UNN | 17.8 | 25.5 | 39.5 |

the observed state remain. The procedure resulted in a relatively large positive ice thickness drift in the Amundsen region. The GRIS LSC model is thus almost insensitive to the basal melting rate anomaly in this region as the error in the grounded ice is too large. Conversely, there are relatively low errors in the Ross region and the response to the oceanic perturbation is stronger there. In addition, in some regions there are compensating errors (positive bias in some places and negative bias in others) which complicate the analysis of the response curve in terms of sea level. In addition, as for the initMIP-Antarctica experiments, a homogeneous sub-shelf basal melting rate was used for each individual IMBIE 2016 basin (Rignot and Mouginot, 2016). The value for each basin was computed as the basin-averaged sub-shelf basal melting rates that ensure a minimal ice shelf thickness Eulerian derivative in a forward experiment with constant climate forcing and a fixed grounding-line position. The spatial average is needed in order to smooth the otherwise noisy melt rates, and it also provides melt rates for a changing ice shelf geometry. Nonetheless, in computing the spatial average the model tends to overestimate the melt away from the grounding line and underestimate it in its vicinity, where it has the largest impact on ice dynamics.

The response of the PISM AWI model is slightly lower than the median response across all models. This can be partially understood by breaking down the response to the indi-

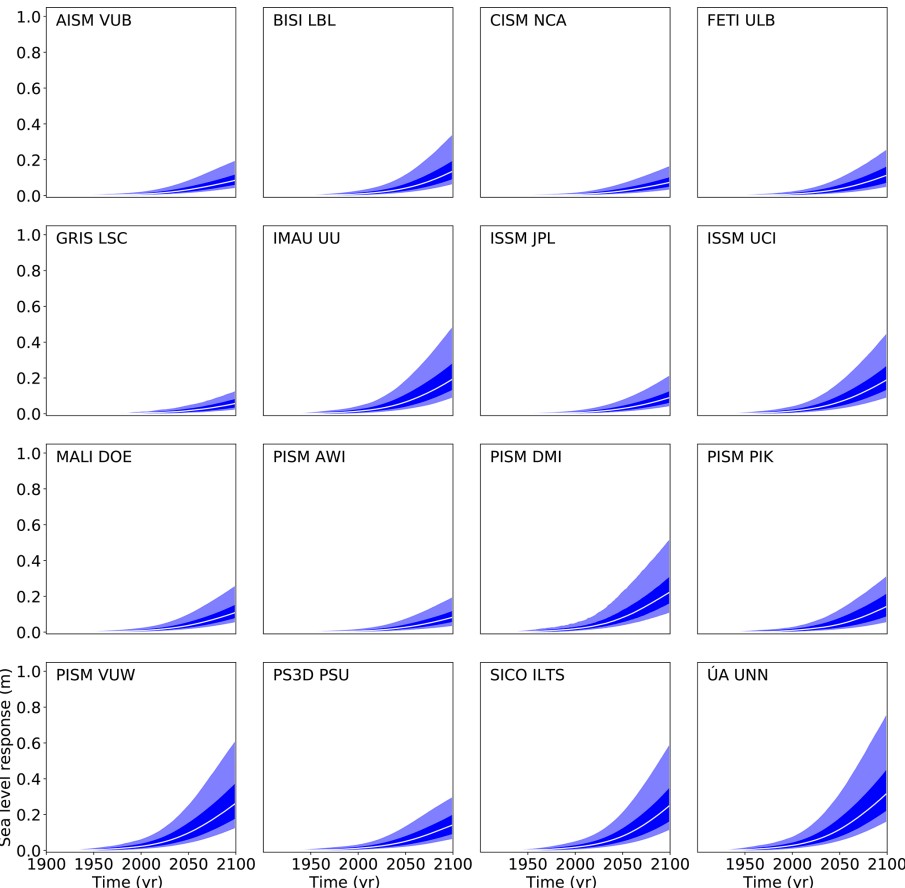

**Figure 8.** Projection of Antarctica's sea level contribution under the RCP2.6 carbon concentration scenario following the procedure depicted in Fig. 1 and detailed in Sect. 2. The white line represents the median value, the dark shading the likely range (66th percentile around the median), and the light shading the very likely range (90th percentile around the median). Compare Tables 7–10 for the values and their comparison to the other scenarios.

vidual sectors and their discrepancy between modelled spin-up and observed state. In the Weddell Sea sector the initial grounding-line position is already retreated inland, lowering this sector's potential sea level contribution in the forward runs. This could explain the relatively low sensitivity to the melt perturbation compared to other models. While the initial grounding line in the Amundsen Sea sector is captured well compared to observations, the corresponding ice shelf area is overestimated, leading to a stronger buttressing and therefore a limited drainage via Pine Island and Thwaites glacier. Basel melt anomalies above $8 \, \mathrm{m \, yr^{-1}}$ are required to eliminate the additional ice shelf area in this region and thus have less influence on the sea level contribution. A small basal melt anomaly of $1 \, \mathrm{m \, yr^{-1}}$ is already sufficient to melt away large portions of the Ronne and Ross Ice Shelf with only a minor impact on the grounding-line position on the timescales considered here.

The sea level contribution from AISM VUB is also somewhat below the median of the 16 state-of-the-art ice sheet models. It has a median of 0.06 m for RCP2.6 (all-model me-

dian mean is 0.13 m) and a median of 0.13 m for RCP8.5 (all-model median mean is 0.17 m). Experiments with the high $16 \, \mathrm{m \, yr^{-1}}$ basal melt anomaly were not performed because most of the ice shelves are lost after 200 years, and the model does not include a proper treatment for calving at a moving margin or for the specific force balance of a calving front at the grounding line. Possible reasons for differences with the other models are most likely the various approximations made to simulate grounding-line mechanics, and it seems fair to state that no model does this perfectly. Additionally, AISM VUB is run at a resolution of 20 km over the entire model domain, which is rather course. Sub-grid-scale mechanisms are not described, and this may affect the model sensitivity. Another difference is that AISM VUB has a freely evolving grounding line in the spin-up at the expense of a slight mismatch of the ice sheet geometry compared to the observations. In contrast to many of the other models, AISM VUB represents the hindcast of the historical contribution to sea level rise very well using the linear response functions.

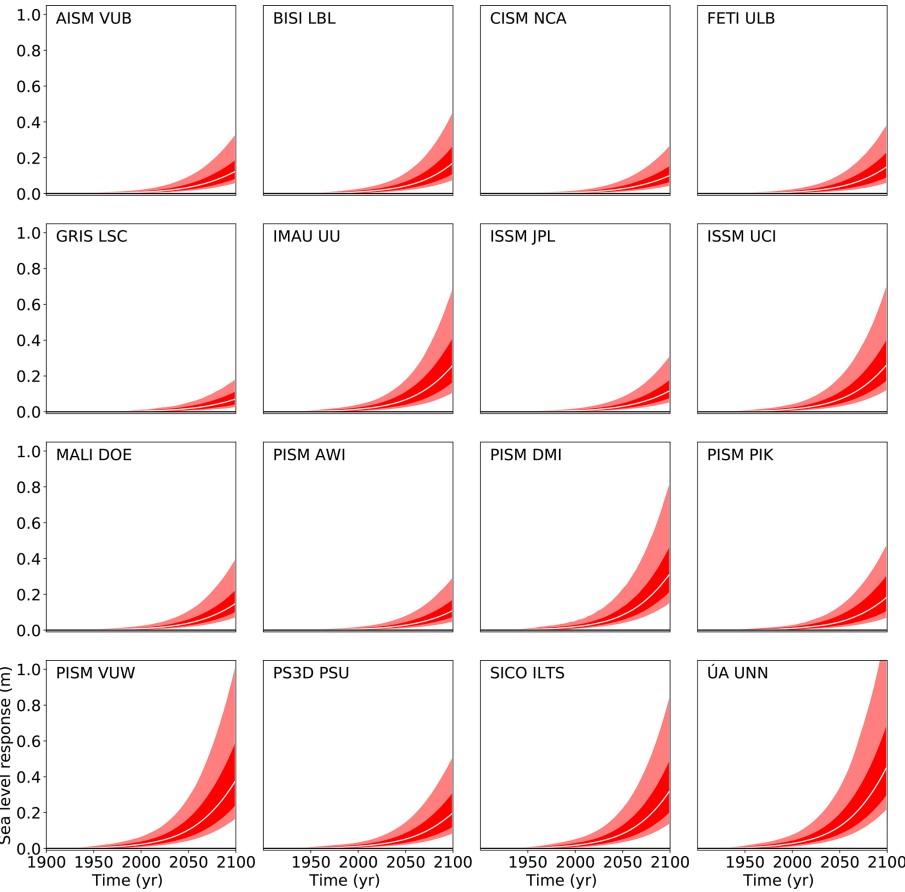

**Figure 9.** Projection of Antarctica's sea level contribution under the RCP8.5 carbon concentration scenario following the procedure depicted in Fig. 1 and detailed in Sect. 2. The white line represents the median value, the dark shading the likely range (66th percentile around the median), and the light shading the very likely range (90th percentile around the median). Compare Tables 7–10 for the values and their comparison to the other scenarios.

Also, CISM NCA is one of the less sensitive models, with median sea level contributions ranging from 0.07 m for RCP2.6 (Table 7) to 0.10 m for RCP8.5 (Table 10) compared to all-model means of 0.13 and 0.17 m, respectively. The largest responses are in the East Antarctic, Weddell, and Ross sectors, with little change in the Amundsen sector and Antarctic Peninsula. One reason for the low sensitivity may be the multi-millennial spin-up procedure, during which the ice was nudged toward present-day thickness by adjusting basal sliding coefficients (beneath grounded ice) and basal melt rates (beneath floating ice). There was no attempt to match recent mass loss, and the spun-up ice sheet has considerable inertia. In multi-century CISM NCA simulations substantial thinning and retreat in the Thwaites basin is seen, driven largely by MISI. The retreat, however, only begins after several decades of increased basal melting. Apart from MISI, CISM NCA, like all models in this intercomparison, has no special mechanisms (e.g. hydrofracture) to promote fast grounding-line retreat.

The ISSM JPL model shows a relatively weak sensitivity to basal ice shelf melt. By comparison the ISSM UCI version of the model shows a medium to strong response. The main difference between the two ISSM models is the different mesh resolution over the ice shelves and especially close to the grounding lines. The ISSM JPL model has a finer resolution over all the ice shelves (less than 5 km for all the floating ice at the beginning of the simulation), while the ISSM UCI models has a slightly coarser resolution in some parts of the ice shelves as the resolution is mostly refined in regions with large gradients of velocities. Otherwise, most parameters and parameterizations are similar, including the friction law and the exclusion of melt on partially grounded cells.

In order to understand the response of FETI ULB in a more global context, especially the relatively weak response in the Amundsen Sea sector compared to other regions such as the Weddell sector, we can add that this is most likely related to underestimating the present-day peak melt rates near the grounding line in this sector. This is due to both the applied spatial resolution and the use of the PICO model (Potsdam

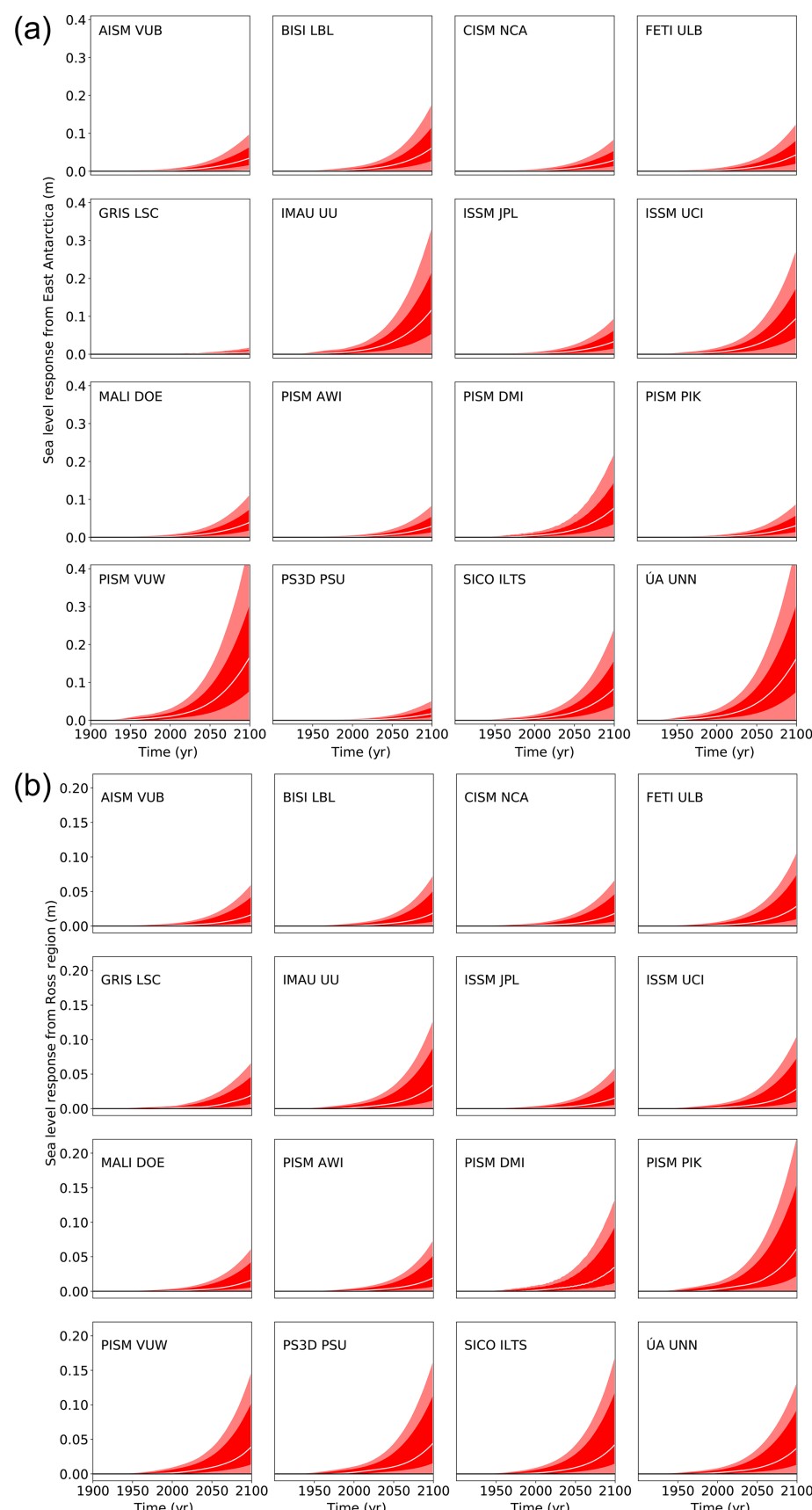

**Figure 10.**

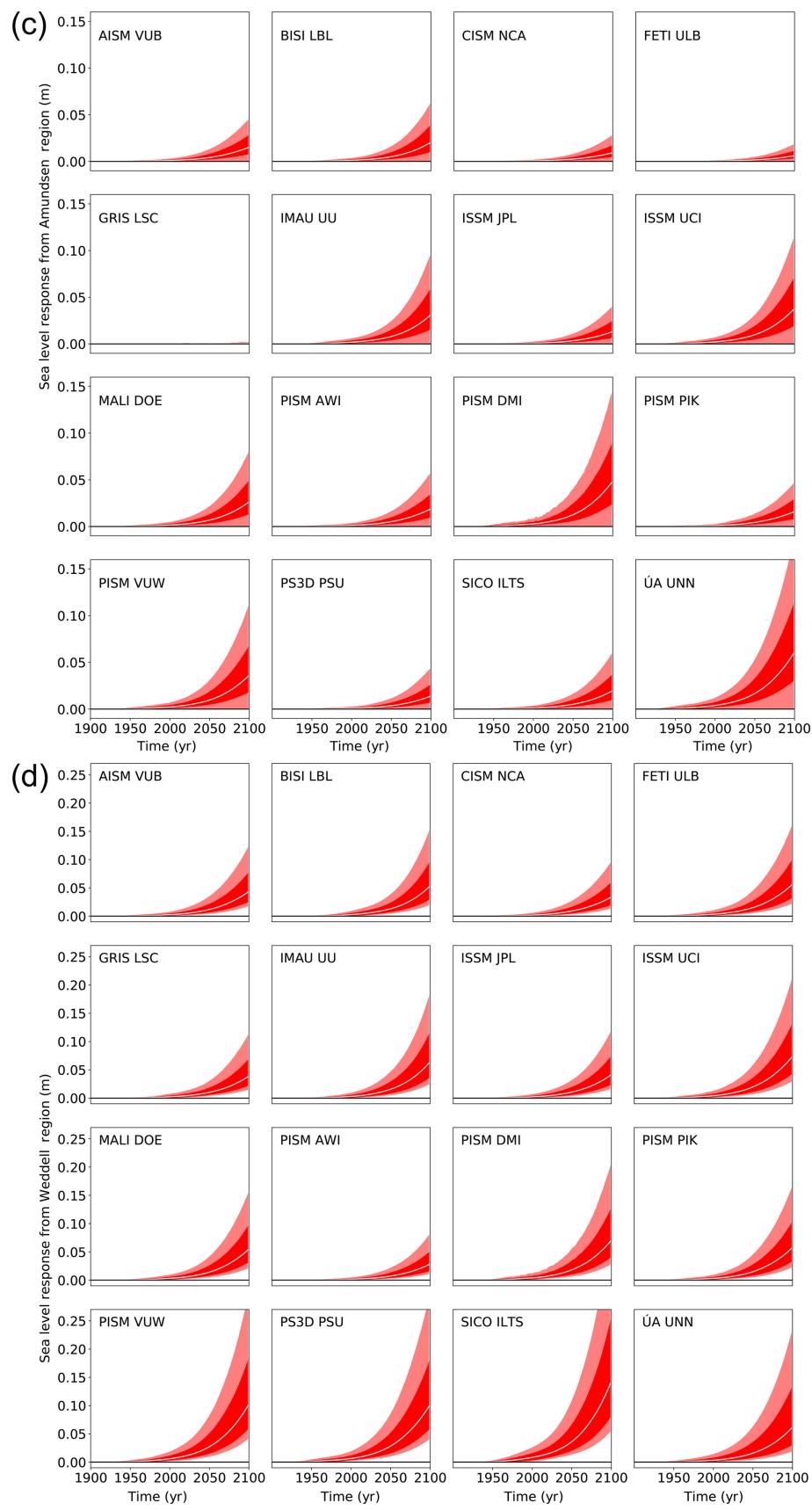

**Figure 10.**

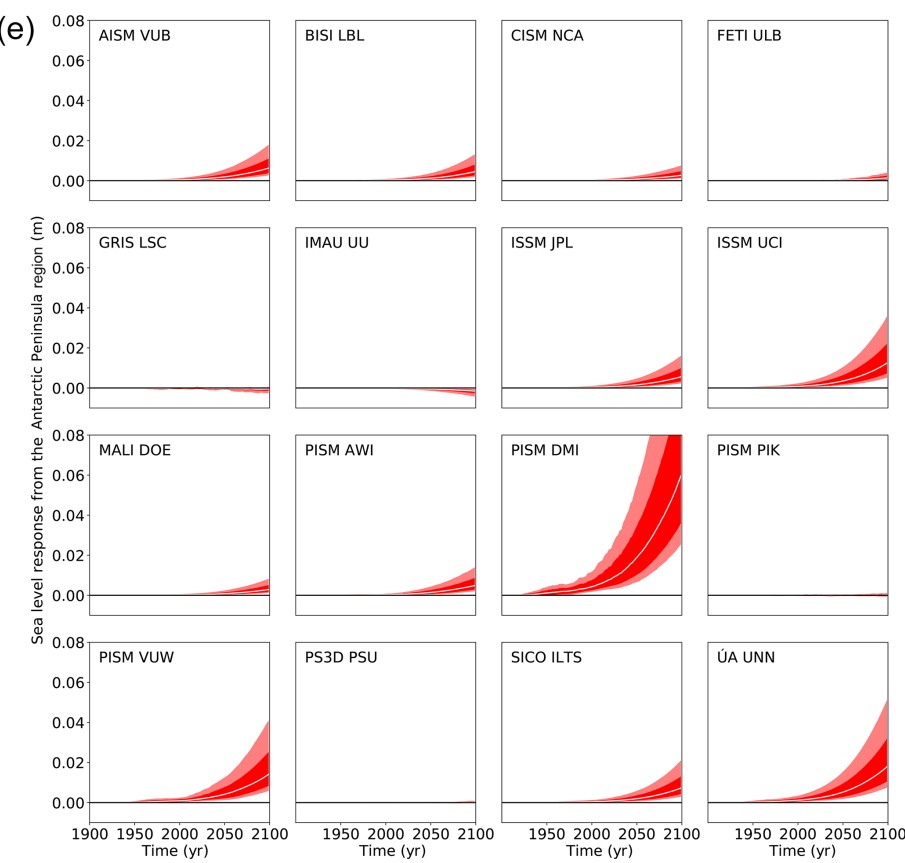

**Figure 10. (a)** Projection of East Antarctica's sea level contribution under the RCP8.5 carbon concentration scenario following the procedure depicted in Fig. 1 and detailed in Sect. 2. The white line represents the median value, the dark shading the likely range (66th percentile around the median), and the light shading the very likely range (90th percentile around the median). **(b)** Projection of the Ross sector's sea level contribution under the RCP8.5 carbon concentration scenario as in panel **(a)**. **(c)** Projection of the Amundsen sector's sea level contribution under the RCP8.5 carbon concentration scenario as in panel **(a)**. **(d)** Projection of the Weddell sector's sea level contribution under the RCP8.5 carbon concentration scenario as in panel **(a)**. **(e)** Projection of the Antarctic Peninsula's sea level contribution under the RCP8.5 carbon concentration scenario as in panel **(a)**.

Ice-shelf Cavity mOdel; Reese et al., 2018b) (and associated temperature and salinity data in front of the ice shelf). For the Amundsen region, the PICO model leads to peak melt rates below $20\,\mathrm{m\,a^{-1}}$, and adding even $32\,\mathrm{m\,a^{-1}}$ to this still remains to the low side compared to observed. Improvements on this sector are currently on the way in order to improve the match with present-day and glacier mass losses. The same applies to the Weddell sector, where sub-shelf melting may be overestimated.

The MALI DOE model response is also overall less sensitive than the model mean. For some Antarctic sectors (e.g. the Ross and Amundsen Sea sectors), average sea level trends or the shape of sea level curves from MALI DOE compare well with those from other higher-order[1] models (e.g.

BISICLES and ISSM UCI). For other sectors (Weddell and EAIS), MALI DOE response functions compare well with only one other model (BISI LBL and SICO ILTS, respectively). In general, there is no obvious correlation between the only two three-dimensional, formally higher-order models in the study (MALI DOE and ISSM UCI), which suggests that something other than model dynamics is responsible for the differences in model response functions seen here (e.g. choice of model physics or model initialization procedure). For the hindcasting experiments, approximately half of the models compare reasonably well with observed trends in Antarctic mass loss, while the other half are biased towards the high side of mass loss (both in terms of mean trends and upper bounds). MALI DOE is within the former group (Fig. 6), initially overestimating then underestimating mass loss trends in the middle part of the observational record, but in good agreement in terms of both the mean trend and range in the latter part of the record. For experiments under the RCP scenarios, 7–10 models are (visu-

---

[1]Here, by higher-order, we mean those that are formally a higher-order approximation of the Stokes equations, e.g. 1st-order (Blatter–Pattyn) or L1L2, as opposed to "hybrid" models which are a more ad hoc approximation to formally higher-order models.

**Table 8.** Percentiles of the probability distribution of the sea level contribution of Antarctica for different ice sheet models under the RCP4.5 climate scenario. The 50th percentile corresponds to the median; 16.6 %–83.3 % is the so-called "likely range" as denoted in the IPCC reports. The "very likely range" is given by 5 %–95 %.

| RCP4.5 | Antarctica sea level contribution percentiles (m) | | | | |
|---|---|---|---|---|---|
| Model | 5 % | 16.6 % | 50 % | 83.3 % | 95 % |
| AISM VUB | 0.05 | 0.07 | 0.10 | 0.14 | 0.24 |
| BISI LBL | 0.07 | 0.10 | 0.15 | 0.22 | 0.39 |
| CISM NCA | 0.04 | 0.05 | 0.08 | 0.12 | 0.19 |
| FETI ULB | 0.05 | 0.08 | 0.13 | 0.19 | 0.30 |
| GRIS LSC | 0.03 | 0.04 | 0.06 | 0.09 | 0.15 |
| IMAU UU | 0.10 | 0.15 | 0.22 | 0.33 | 0.56 |
| ISSM JPL | 0.05 | 0.07 | 0.10 | 0.14 | 0.25 |
| ISSM UCI | 0.10 | 0.15 | 0.22 | 0.32 | 0.55 |
| MALI DOE | 0.06 | 0.09 | 0.12 | 0.18 | 0.31 |
| PISM AWI | 0.04 | 0.06 | 0.09 | 0.14 | 0.23 |
| PISM DMI | 0.13 | 0.18 | 0.26 | 0.36 | 0.64 |
| PISM PIK | 0.06 | 0.10 | 0.16 | 0.25 | 0.37 |
| PISM VUW | 0.14 | 0.20 | 0.30 | 0.45 | 0.76 |
| PS3D PSU | 0.07 | 0.10 | 0.16 | 0.24 | 0.37 |
| SICO ILTS | 0.12 | 0.18 | 0.28 | 0.40 | 0.69 |
| ÚA UNN | 0.18 | 0.25 | 0.37 | 0.53 | 0.90 |
| All models | 0.05 | 0.08 | 0.15 | 0.31 | 0.47 |

**Table 10.** Percentiles of the probability distribution of the sea level contribution of Antarctica for different ice sheet models under the RCP8.5 climate scenario. The 50th percentile corresponds to the median; 16.6 %–83.3 % is the so-called "likely range" as denoted in the IPCC reports. The "very likely range" is given by the 5th–95th percentiles.

| RCP8.5 | Antarctica sea level contribution percentiles (m) | | | | |
|---|---|---|---|---|---|
| Model | 5 % | 16.6 % | 50 % | 83.3 % | 95 % |
| AISM VUB | 0.06 | 0.08 | 0.13 | 0.19 | 0.33 |
| BISI LBL | 0.08 | 0.11 | 0.17 | 0.27 | 0.46 |
| CISM NCA | 0.04 | 0.06 | 0.10 | 0.16 | 0.27 |
| FETI ULB | 0.06 | 0.09 | 0.15 | 0.23 | 0.39 |
| GRIS LSC | 0.03 | 0.04 | 0.07 | 0.11 | 0.18 |
| IMAU UU | 0.11 | 0.17 | 0.26 | 0.42 | 0.70 |
| ISSM JPL | 0.05 | 0.08 | 0.12 | 0.18 | 0.31 |
| ISSM UCI | 0.12 | 0.18 | 0.27 | 0.41 | 0.71 |
| MALI DOE | 0.07 | 0.10 | 0.15 | 0.23 | 0.40 |
| PISM AWI | 0.05 | 0.07 | 0.11 | 0.17 | 0.30 |
| PISM DMI | 0.15 | 0.22 | 0.33 | 0.47 | 0.83 |
| PISM PIK | 0.07 | 0.11 | 0.19 | 0.31 | 0.48 |
| PISM VUW | 0.17 | 0.24 | 0.38 | 0.60 | 1.03 |
| PS3D PSU | 0.08 | 0.12 | 0.20 | 0.31 | 0.51 |
| SICO ILTS | 0.14 | 0.20 | 0.33 | 0.50 | 0.86 |
| ÚA UNN | 0.22 | 0.30 | 0.46 | 0.70 | 1.25 |
| All models | 0.06 | 0.09 | 0.18 | 0.38 | 0.61 |

**Table 9.** Percentiles of the probability distribution of the sea level contribution of Antarctica for different ice sheet models under the RCP6.0 climate scenario. The 50th percentile corresponds to the median; 16.6 %–83.3 % is the so-called "likely range" as denoted in the IPCC reports. The "very likely range" is given by the 5th–95th percentiles.

| RCP6.0 | Antarctica sea level contribution percentiles (m) | | | | |
|---|---|---|---|---|---|
| Model | 5 % | 16.6 % | 50 % | 83.3 % | 95 % |
| AISM VUB | 0.05 | 0.07 | 0.10 | 0.14 | 0.25 |
| BISI LBL | 0.07 | 0.10 | 0.14 | 0.22 | 0.37 |
| CISM NCA | 0.04 | 0.05 | 0.08 | 0.12 | 0.20 |
| FETI ULB | 0.05 | 0.08 | 0.13 | 0.19 | 0.30 |
| GRIS LSC | 0.03 | 0.04 | 0.06 | 0.09 | 0.15 |
| IMAU UU | 0.10 | 0.14 | 0.22 | 0.33 | 0.53 |
| ISSM JPL | 0.05 | 0.07 | 0.10 | 0.14 | 0.25 |
| ISSM UCI | 0.10 | 0.15 | 0.22 | 0.32 | 0.54 |
| MALI DOE | 0.06 | 0.09 | 0.12 | 0.18 | 0.31 |
| PISM AWI | 0.04 | 0.06 | 0.09 | 0.13 | 0.22 |
| PISM DMI | 0.13 | 0.18 | 0.26 | 0.36 | 0.64 |
| PISM PIK | 0.06 | 0.09 | 0.16 | 0.25 | 0.37 |
| PISM VUW | 0.14 | 0.20 | 0.30 | 0.45 | 0.77 |
| PS3D PSU | 0.07 | 0.10 | 0.16 | 0.24 | 0.38 |
| SICO ILTS | 0.12 | 0.18 | 0.27 | 0.40 | 0.67 |
| ÚA UNN | 0.18 | 0.25 | 0.37 | 0.54 | 0.93 |
| All models | 0.05 | 0.08 | 0.15 | 0.31 | 0.47 |

ally) in agreement regarding the mean and bounds on future sea level rise from Antarctica for most of the Antarctic sectors investigated. MALI DOE is generally within this group. An exception to this can be found for the Amundsen Sea sector (ASE) under RCP8.5, in which MALI DOE is slightly on the high side (for both the mean trends and upper bounds) relative to other models. This is likely an expression of MALI DOE already exhibiting a trend towards significant mass loss from the ASE in its unperturbed initial condition (see e.g. Fig. 4a in Seroussi et al., 2019). While this trend is largely removed by differencing with the control run, the unstable and non-linear nature of the retreat in ASE likely increases the magnitude of mass loss forced under RCP scenarios.

The strongly sub-linear response of PISM PIK to the additional basal melt forcing in the Ross region (Fig. 4b) is likely to result partially from the applied spin-up procedure. In order to best match present-day observations in the most sensitive part of the West Antarctic Ice Sheet (mainly the Amundsen Sea sector) and due to computational costs, PISM PIK was initialized with a transient spin-up at the end of a 600-year run forced by present-day climatic boundary conditions that is not in equilibrium. This allows us to reproduce recent change rates in the Amundsen sector, but trends in other regions (e.g. Siple Coast) can exceed present rates and superpose the ice sheet's response to the forcing in the experiments. Another reason for PISM PIK's sub-linear behaviour

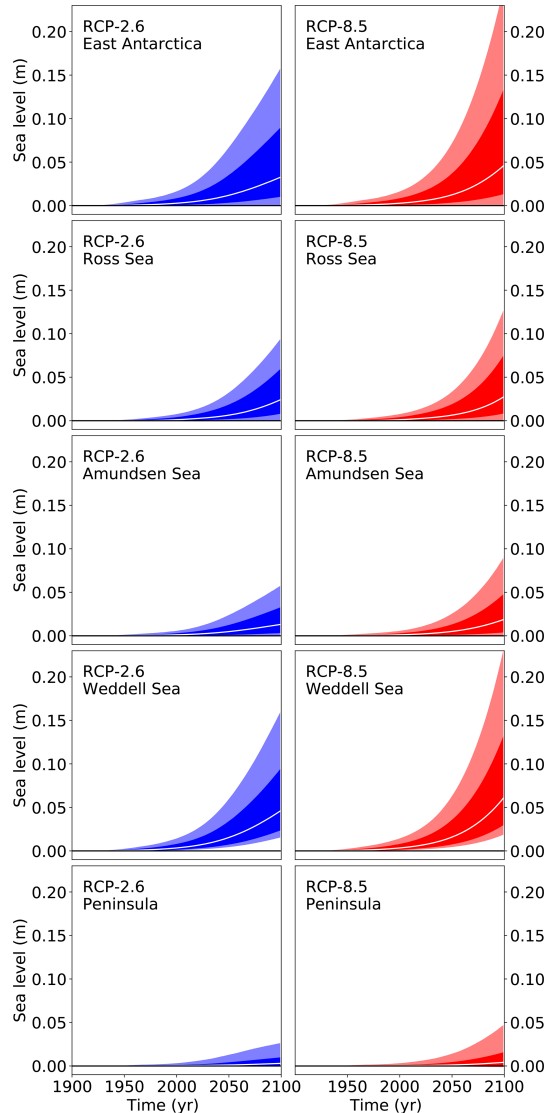

**Figure 11.** Projections from all models of the future sea level contribution of the different Antarctic sectors following the procedure depicted in Fig. 1 and detailed in Sect. 2. The white line represents the median value, the dark shading the likely range (66th percentile around the median), and the light shading the very likely range (90th percentile around the median).

in this region is likely related to ice shelf break-up in the forcing experiments as a result of the interplay of strong melting near the grounding line and the applied calving mechanism ("eigencalving"; Levermann et al., 2012). For all additional sub-shelf melt rates, disintegration of the Ross Ice Shelf is initiated near the grounding line, which is rather unrealistic. As the ice shelf remainders exert almost no buttressing onto the grounded ice stream flow, there is almost no response of the sea-level-relevant ice volume to the magnitude of melt forcing applied. The peninsula region, where in the initial state almost no ice shelves are present, also shows no signif-

icant response. In contrast to the Ross region and peninsula, PISM PIK shows slightly super-linear scaling in the Weddell region. Although large portions of the Filchner–Ronne Ice Shelf melt and calve off, a small part pinned to the Korff and Henry ice rises remains in the forcing experiments and exerts buttressing. For higher melt rates, those ice shelf remainders exert decreasing buttressing on grounded ice sheet flow, which is associated with enhanced sea level contributions.

A number of models scatter quite closely around the median model sensitivity when measured in the total ice sheet response. In general, the BISICLES model falls within the median group of models but is generally on the more responsive side in that grouping. This has much to do with the initialization of the model – the BISICLES runs use the initial condition from the initMIP exercise; since BISICLES falls in the median group of models in that exercise, it is unsurprising that it also falls in the median group in this context. It is well known that models with insufficient resolution will generally tend to underestimate marine ice sheet response, which is borne out in the results here, in which the coarser-resolution (1 km resolution) BISICLES is less responsive than the finer-resolution runs (500 m); however, the differences are small in this case because both cases are sufficiently resolved to capture the dynamics in play. The BISICLES response function appears to be on the higher side in regions with no appreciable present-day grounding-line retreat (like East Antarctica), possibly because it is able to better capture the onset of new retreat and deploy sufficient resolution there. BISICLES appears more in line with other models for the Amundsen Sea sector, possibly because substantial retreat and loss is already underway in that sector, so the actual dynamics remain relatively unchanged with increased sub-shelf melting.

One possibility distinguishing the PS3D PSU model is the boundary layer parameterization of ice flux across grounding lines (Schoof, 2007) imposed as a condition on ice velocity across the grounding line. This enables grounding-line migration to be simulated reasonably well without much higher grid resolution. The sub-grid grounding-line treatment also ensures no substantial oceanic melt upstream of the grounding line, in contrast to some models. Note that the recently proposed mechanisms of hydrofracturing by surface meltwater and structural failure of large ice cliffs (DeConto and Pollard, 2016) are not enabled for the LARMIP experiments. Without these, previous studies with this model have found little retreat in East Antarctic basins with moderate sub-ice ocean melting alone, consistent with the generally smaller response for East Antarctica in the experiments here.

Similarly to the less sensitive models, the models which have an overall stronger response to basal ice shelf melting have similarly diverse reasons for their dynamics. The sea level contribution of IMAU UU is in the upper half of the ensemble and most similar to the ISSM UCI model as well as to some extent PISM DMI and PISM VUW (Figs. 6, 8, 9, 10a–d, Tables 6–10). The only exception is the Antarc-

tic Peninsula region (Fig. 10e), where IMAU UU CE2 has the lowest response in the ensemble. This may be attributed to the relatively low horizontal grid resolution of the model ($32 \times 32$ km), which prevents the resolution of small-scale features important for this region.

Compared to other models in the ensemble, PISM VUW yields a hindcast sea level contribution that is above observed values and a large spread in the projected contribution by 2100 under RCP8.5 (less than 0.2 m up to ca. 1 m). Model differences appear to be spatially variable. For example, PISM VUW projects SL contributions from the Ross Sea sector that are very consistent with most other models in the ensemble, as well as contributions from the Antarctic Peninsula and from the Amundsen Sea embayment that lie consistently between the highest and lowest models. It is primarily the contributions from the Weddell Sea sector that are very much higher than most other models, and since the combined East Antarctic contributions from PISM VUW appear to be similarly skewed, it is reasonable to infer that the large Weddell Sea contributions are principally sourced from the Recovery Basin in the eastern part of the sector. It seems reasonable that the different modelled response here arises from the way in which basal conditions are parameterized: for example, the basal substrate yield strength that determines the propensity of ice in this area to stream.

The set-up used for the LARMIP simulations with SICO ILTS is the same as that used for the ISMIP6 projections (http://tinyurl.com/ismip6-wiki-ais, last access: 6 January 2020; publication in preparation; see also Appendix A15). In most regions, the results show a rather high sensitivity to the applied ice shelf basal melting anomalies. This is probably because any grid cell for which the centre point is floating is assigned the full ice shelf basal melting rate, even if the grid cell is near the grounding line and parts of it might be detectable as grounded via a sub-grid interpolation technique. Relative to the other models, the sensitivity is largest for the Ross region. This correlates with the fact that for this region, in particular the West Antarctic part including the Siple Coast ice streams, the regional basal sliding inversion (Appendix A15) produces the largest values for the basal sliding coefficient.

Out of all the models the ÚA UNN model has the overall highest projection for future sea level contribution for Antarctica as a whole. This can mostly be attributed to the high amount of future sea level rise the model projects for East Antarctica as this region is the single largest contributor to future sea level rise. The model also projects a relatively strong contribution from the Amundsen and Antarctic Peninsula sectors, with more average projections when compared to the rest of the model ensemble from the Ross and Weddell sector. One possible explanation for this is that the ÚA UNN model overestimates the past Antarctic sea level rise when compared to observations (Fig. 6). This would likely predispose the model to have a relatively large projection for the

**Table 11.** Sea level contributions from basal ice shelf melting from Antarctica within the 21st century from all models for the different emission scenarios in metres. TS6

| Scenario | 5 % | 16.6 % | 50 % | 83.3 % | 95 % |
|---|---|---|---|---|---|
| RCP2.6 | 0.05 | 0.07 | 0.14 | 0.27 | 0.40 |
| RCP4.5 | 0.05 | 0.08 | 0.15 | 0.31 | 0.47 |
| RCP6.0 | 0.05 | 0.08 | 0.15 | 0.31 | 0.47 |
| RCP8.5 | 0.06 | 0.09 | 0.18 | 0.38 | 0.61 |

**Table 12.** Sea level rate contributions from basal ice shelf melting from Antarctica in 2100 from all models for the different emission scenarios in millimetres per year or centimetres per decade.

| Scenario | 5 % | 16.6 % | 50 % | 83.3 % | 95 % |
|---|---|---|---|---|---|
| RCP2.6 | 0.8 | 1.2 | 2.2 | 4.2 | 6.3 |
| RCP4.5 | 1.0 | 1.6 | 2.9 | 5.6 | 8.5 |
| RCP6.0 | 1.1 | 1.7 | 3.1 | 6.1 | 9.2 |
| RCP8.5 | 1.5 | 2.3 | 4.4 | 8.9 | 14.0 |

future contribution of Antarctic sea level rise when compared to other models that more accurately match the hindcast.

Across all models one can say that even though the temperature difference between the scenarios is significant, the difference in the Antarctic ice sheet response is existent but percentage-wise smaller. Table 11 gives a summary across the scenarios for all ice sheet models combined. The corresponding time series are given in Fig. 12. The relative warming difference between RCP8.5 and RCP2.6 within this century (according to the median values) is about $(3.7–1.0\,\mathrm{K}) / 1.0\,\mathrm{K} = 270\,\%$ (Stocker et al., 2013). For comparison the Antarctic sea level contribution is (according to Table 11) about $(0.18–0.14\,\mathrm{m}) / 0.14\,\mathrm{m} = 29\,\%$ TS4. One reason for this is the time delay between the surface forcing and the subsurface oceanic forcing that is experienced by the ice shelves. The relative difference in global mean temperature increase between the scenarios also increases with time during this century. However, the strongly reduced relative sea level difference between the scenarios mainly reflects the inertia in the ice sheet dynamics, which responds to the forcing in a time-delayed way as can be seen from the response functions in Fig. 5a–e. For the upper end of the very likely range (95th percentile) this ratio is larger at $(0.61–0.40\,\mathrm{m}) / 0.40\,\mathrm{m} = 53\,\%$ TS5 but still lower than the scenario ratio of the warming. This does not hold for the rate of change in sea level (Fig. 13, Table 12), which is $(4.4–2.2\,\mathrm{mm\,yr^{-1}}) / 2.2\,\mathrm{mm\,yr^{-1}} = 100\,\%$.

Due to increasing interest of society and by extension of the Intergovernmental Panel on Climate Change, we also provide the sea level contributions of Antarctica due to basal ice shelf melting until the middle of the 21st century (Table 13), the associated rate of sea level contribution (Table 14), and the contribution of Antarctica within the next

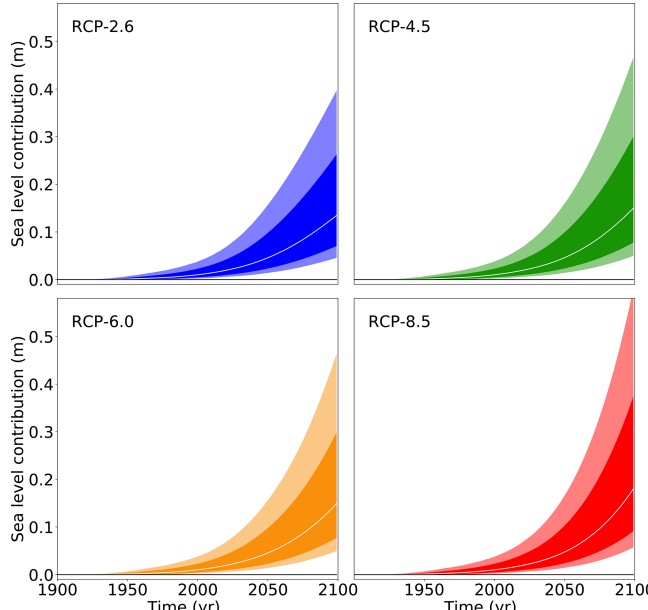

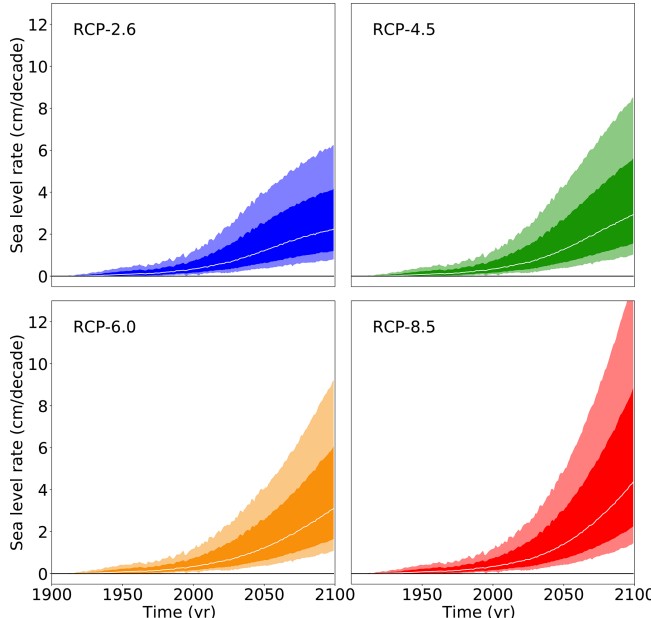

**Figure 12.** Projections from all models of the future sea level contribution of the Antarctic ice sheet under different atmospheric carbon concentration scenarios following the procedure depicted in Fig. 1 and detailed in Sect. 2. The white line represents the median value, the dark shading the likely range (66th percentile around the median), and the light shading the very likely range (90th percentile around the median).

**Figure 13.** Projections from all models of the rate of future sea level contribution of the Antarctic ice sheet under different atmospheric carbon concentration scenarios following the procedure depicted in Fig. 1 and detailed in Sect. 2. The white line represents the median value, the dark shading the likely range (66th percentile around the median), and the light shading the very likely range (90th percentile around the median).

**Table 13.** Sea level contributions from basal ice shelf melting from Antarctica until the middle of the 21st century (year 2050) from all models for the different emission scenarios in metres. The projections are scenario independent until 2050, with the uncertainty of the forcing and across ice sheet models determining the uncertainty in the future.

| Scenario | 5 % | 16.6 % | 50 % | 83.3 % | 95 % |
|----------|-----|--------|------|--------|------|
| RCP2.6   | 0.01 | 0.02 | 0.05 | 0.10 | 0.16 |
| RCP4.5   | 0.01 | 0.02 | 0.05 | 0.10 | 0.16 |
| RCP6.0   | 0.01 | 0.02 | 0.05 | 0.10 | 0.16 |
| RCP8.5   | 0.01 | 0.02 | 0.05 | 0.10 | 0.17 |

**Table 14.** Sea level rate contributions from basal ice shelf melting from Antarctica in the middle of the 21st century (year 2050) from all models for the different emission scenarios in metres. Different from the sea level rise, the rate of sea level rise already depends on the scenario even in the first half of the century. Note that in RCP4.5, although it ends up lower in radiative forcing in the year 2100, it rises more quickly in the beginning of the century, which leads to a slightly higher contribution of Antarctica to the rate of sea level rise until 2050 compared to RCP6.0.

| Scenario | 5 % | 16.6 % | 50 % | 83.3 % | 95 % |
|----------|-----|--------|------|--------|------|
| RCP2.6   | 0.4 | 0.6 | 1.2 | 2.5 | 3.9 |
| RCP4.5   | 0.4 | 0.7 | 1.3 | 2.7 | 4.3 |
| RCP6.0   | 0.4 | 0.7 | 1.2 | 2.5 | 4.0 |
| RCP8.5   | 0.4 | 0.7 | 1.4 | 3.0 | 4.9 |

## 5 Discussion and conclusions

The projections of the Antarctic contribution to future sea level rise have to be seen in comparison with other studies. The fifth assessment report of the Intergovernmental Panel on Climate Change (IPCC AR5) only had limited process-based model simulations available (Gladstone et al., 2012) and thus estimated a likely range for the ice dynamical contribution of the Antarctic ice sheet of −1 to 16 cm (Church et al., 2013). This estimate was largely based on statistical con-

30 years (Table 15). There is practically no scenario dependence in these numbers. This is to be expected since the global warming signal only differs significantly between scenarios for time periods beyond 2040. It is found that in the next 30 years the median contribution of Antarctica to global sea level rise from basal ice shelf melting is 3 cm, with a likely range between 1 and 6 cm. The applicability of these numbers is strongly limited by the caveats of the method as described in the next section.

**Table 15.** Sea level contributions from basal ice shelf melting from Antarctica between 2020 and 2050.

| Scenario | 5 % | 16.6 % | 50 % | 83.3 % | 95 % |
|---|---|---|---|---|---|
| RCP2.6 | 0.01 | 0.01 | 0.03 | 0.05 | 0.09 |
| RCP4.5 | 0.01 | 0.01 | 0.03 | 0.06 | 0.09 |
| RCP6.0 | 0.01 | 0.01 | 0.03 | 0.05 | 0.09 |
| RCP8.5 | 0.01 | 0.01 | 0.03 | 0.06 | 0.10 |

siderations (Little et al., 2013a, b) which do not represent a response to future warming but merely estimate the possible statistical range of responses based on variations in observed discharge velocities. Thus, these estimates are scenario independent as was the projection by the IPCC AR5. The IPCC AR5, however, added a footnote saying that the likely range could increase by "several decimetres" if the West Antarctic Ice Sheet becomes unstable. The following special reports of the IPCC that addressed sea level rise (Masson-Delmotte et al., 2018; Pörtner et al., 2019) included the estimates obtained with the same procedure applied here but for earlier ice sheet models (Levermann et al., 2014). A mere extrapolation of observed ice dynamic contributions from Antarctica constrained by its future sea level commitment yields a likely range of 9 to 19 cm for the end of the 21st century under RCP8.5 (Mengel et al., 2016). Similar values are obtained with more elaborated statistical methods (Kopp et al., 2014, 2017).

In the meantime a number of other studies have shed light on the importance of process-based projections of Antarctica (e.g. Arthern et al., 2015; Favier et al., 2014; Gong et al., 2014; Joughin et al., 2014). For example, it was shown that feedbacks between the Antarctic ice sheet and the surrounding ocean and atmosphere can strongly increase the ice loss from Antarctica (Golledge et al., 2019). In a model (Pollard and DeConto, 2009) that was able to reproduce paleo-evidence of the grounding-line retreat in a location in Antarctica (Naish et al., 2009), it was shown that the inclusion of additional physical surface processes (Pollard et al., 2015) yields more than a doubling of the previous high estimate of the ice loss considered possible from Antarctica (DeConto and Pollard, 2016). Although it was shown that the paleo-constraints used in these simulations were insufficient to properly constrain future projections (Edwards et al., 2019), it cannot be ruled out that these processes are significant. Consequently there is large uncertainty in the ice sheet community regarding the possible contribution of Antarctica to future sea level rise, as can be seen from two separate expert elicitations before (Bamber and Aspinall, 2013) and after the IPCC AR5 (Bamber et al., 2019). These expert elicitations include all known and unknown uncertainties of possible responses of the Antarctic ice sheet to future warming, and thus the likely and in particular the very likely ranges found in the elicitations are wider than those found with the procedure de-

scribed here. By comparison the likely and very likely ranges found in the earlier estimate based on the linear response theory are the largest ranges if no additional processes such as hydrofracturing and cliff calving are included.

The latest assessment of the Intergovernmental Panel on Climate Change based on the literature published after IPCC AR5 was carried out in the special report on the ocean and cryosphere (Oppenheimer et al., 2019). The author team estimated the Antarctic contribution within the 21st century under the RCP8.5 scenario to be 10 cm, with a likely range of 2 to 23 cm. In this study the likely range for RCP8.5 of 9 to 38 cm [TS7] is slightly higher compared to the earlier studies with a likely range of 4 to 21 cm (Table 6 in Levermann et al., 2014, "Shelf models with time delay"). The same is true for the very likely range over which the current study finds 6 to 61 cm [TS8], while the study with only three ice sheet models found 1 to 37 cm. The 2014 study even found a lower estimate for the very likely range if the time delay was omitted and the atmospheric warming was translated immediately into a scaled subsurface ocean warming (very likely range between 0.04 and 0.43 cm [TS9]). The median estimates in both studies are also very different, with 9 cm in the 2014 and 18 cm [TS10] in the present study for RCP8.5.

The projections of the Antarctic ice sheet's mass loss presented here have strong limitations. First of all they represent only the contribution from basal ice shelf melt. Any calving that might be incorporated in the modelling does not reflect atmospheric or even specific oceanic processes that may enhance calving in a warming world. Hydrofracturing and cliff calving are not explicitly accounted for. The approach neglects a number of processes such as surface-mass-balance-related contributions and mechanisms. There is no mass gain due to additional snowfall or any responses to such a mass addition. In assuming linear response theory, we are able to capture complex temporal responses of the ice sheets, but we neglect any self-dampening or self-amplifying processes. This is particularly relevant in situations in which an instability is dominating the ice loss. This is particularly important for the Marine Ice Sheet Instability (Pattyn et al., 2012; Pattyn and Durand, 2013; Weertman, 1974) that might have already been triggered in the Amundsen Sea sector (Favier et al., 2014; Joughin et al., 2014; Rignot et al., 2014) and might lead to the eventual discharge of the entire marine ice sheet in West Antarctica over a multi-centennial to multi-millennial timescale (Feldmann and Levermann, 2015). The results obtained here are thus relevant, in particular wherever the ice loss is dominated by the forcing as opposed to an internal instability, for example in strong warming scenarios. The study also does not include any feedbacks between the ice sheet and its surroundings. Although feedbacks between the surface mass balance and the ice dynamics are expected to be small (Cornford et al., 2015) there might be significant feedbacks with the ocean circulation both locally and globally (Golledge et al., 2019; Swingedouw et al., 2008). Basal melt rate anomalies are added to the background run of the differ-

ent ice sheet models. However, as melting parameterizations in ice sheet models vary, the sub-shelf melt rates respond differently to the evolving geometry. This is a feedback that is captured in the approach but might be quite different across the models.

These strong caveats that are associated with the approach presented here may either lead to an overestimation or an underestimation of the ice loss from basal ice shelf melting compared to what might occur in reality. In any case the median contribution from basal ice shelf melting of Antarctica under any scenario is found to be higher within the 21st century than it was in the last century. The values obtained here for the basal ice shelf contribution from Antarctica are slightly larger than other probabilistic estimates of the ice loss with (Bakker et al., 2017; Ruckert et al., 2017) and without climate change (Little et al., 2013b). They are much lower than the values that may be obtained if additional processes such as the marine cliff instability and hydrofracturing are included (DeConto and Pollard, 2016). Whether these high estimates, however, can be well constrained by paleo-evidence is still under intense debate (Edwards et al., 2019).

However, due to the very large potential sea level contribution of Antarctica and its high sea level commitment compared to the other contributions (Levermann et al., 2013), the rate of change increases strongly over the century. Under the RCP8.5 scenario the median rate of sea level contribution by the end of the 21st century from basal-melt-induced ice loss from Antarctica alone is with $4.1 \, \mathrm{mm \, yr^{-1}}$ larger than the mean rate of sea level rise observed at the beginning of this century (Dangendorf et al., 2019; Hay et al., 2015; Oppenheimer et al., 2019).

Although the method described here has a large number of caveats it provides an estimate of the role of the uncertainty in the oceanic forcing for the uncertainty in Antarctica's future contribution to sea level rise. By comparison with the earlier study using the same method but only three ice sheet models of an earlier model generation, we find a shift of the sea level contribution to higher values and an increase in the ranges of uncertainty. We thus have to conclude that uncertainty with respect to the ice dynamic contribution of Antarctica due to future warming is still increasing and thus that coastal planning has to take into account that multi-decadal sea level projections are likely to change with an increasing understanding of the ice dynamics and their representation in ice sheet models. This study provides an estimate of the uncertainty in the future contribution of Antarctica to global sea level rise only based on known ice dynamics but including the full range of forcing uncertainty. It substantiates the result of the previous study that Antarctica can become the largest contributor to global sea level rise in the future, in particular if carbon emissions are not abated.

## Appendix A: Brief description of ice sheet models

The model initialization was carried out according to the init-MIP protocol and is described together with the models and their set-up in Seroussi et al. (2019).

### A1 AISM VUB: Antarctic Ice Sheet Model – VUB (Vrije Universiteit Brussel)

The Antarctic ice sheet model AISM VUB derives from a coarse-resolution version used mainly in simulations of the glacial cycles (Huybrechts, 1990, 2002). The version used here is identical to the VUB AISMPALEO model participating in initMIP-Antarctica (Seroussi et al., 2019). It considers thermomechanically coupled flow in both the ice sheet and the ice shelf using the respective shallow-ice approximation and shallow-ice-shelf approximation coupled across a one-grid-cell-wide transition zone. Basal sliding is calculated using a Weertman relation inversely proportional to the height above buoyancy wherever the ice is at the pressure melting point. The horizontal resolution is 20 km and there are 31 layers in the vertical. The model is initialized with a freely evolving geometry until steady state is reached using observed climatologies for the surface mass balance. The sub-shelf basal melt rate is parameterized as a function of local mid-depth (485–700 m) ocean water temperature above the freezing point (Beckmann and Goosse, 2003). A distinction is made between protected ice shelves (Ross and Filchner–Ronne) with a low melt factor and all other ice shelves with a higher melt factor. Ocean temperatures are derived from the LOVECLIM climate model (Goelzer et al., 2016) and parameters are chosen to reproduce observed average melt rates (Depoorter et al., 2013). Heat conduction is calculated in a slab bedrock of 4 km thick underneath the ice sheet. Isostatic compensation is based on an elastic lithosphere floating on a viscous asthenosphere (ELRA model), but this feature is not allowed to evolve further in the current experiments. The LARMIP basal melting rates are applied on top of the present-day melt rates used for the initialization.

### A2 BISI LBL: BISICLES

The finite-volume BISICLES model (Cornford et al., 2013) is used with a modified L1L2 scheme (Schoof and Hindmarsh, 2010) over the entire Antarctic ice sheet. The model employs adaptive mesh refinement (AMR) to vary resolution between a finest resolution (either 1000 or 500 m, depending on the run) near grounding lines and shear margins and 8 km in the interior of the domain. Basal sliding follows a Coulomb-limited friction law (Tsai et al., 2015), resulting in power-law sliding (with a spatially varying friction coefficient) across the majority of the ice sheet with and Coulomb sliding in regions close to flotation. Ice viscosity is computed following Cuffey and Paterson (2010), assuming a prescribed temperature and an enhancement factor. The basal friction coefficient and the enhancement factor are chosen to best match observed surface velocity (Rignot et al., 2011) using a gradient-based, Tikhonov-regularized optimization scheme (Cornford et al., 2015). The grounding-line position is determined using hydrostatic equilibrium, with sub-cell treatment of the friction and a modified driving stress (Cornford et al., 2016). The melt rate is applied only for fully floating cells (as in Seroussi and Morlighem, 2018) and is composed of a base rate and the anomalies specified in the individual experiments. The base melt rate is time varying and designed to prevent ice shelf thickening but permit thinning where flux divergence in the shelf is positive. The surface mass balance is from Arthern et al. (2006). The ice front position is fixed at the extent of the present-day ice sheet. After initialization, the model is relaxed for 2 years, with the base melt rate only applied. For more details on the model and the initialization procedure, we refer to Cornford et al. (2015).

### A3 CISM NCA: Community Ice Sheet Model – NCAR

For LARMIP, the Community Ice Sheet Model (Lipscomb et al., 2019) uses finite-element methods to solve a depth-integrated higher-order approximation (Goldberg, 2011) over the entire Antarctic ice sheet. The model uses a structured rectangular grid with a uniform horizontal resolution of 4 km and five vertical $\sigma$-coordinate levels. The ice sheet is initialized with present-day geometry and an idealized temperature profile, then spun up for 30 000 years using 1979–2016 climatological surface mass balance and surface air temperature from RACMO2 (Lenaerts et al., 2012; van Wessem et al., 2018). During the spin-up, basal friction parameters (for grounded ice) and sub-shelf melt rates (for floating ice) are adjusted to nudge the ice thickness during present-day observations. This method is a hybrid approach between assimilation and spin-up, similar to that described by Pollard and DeConto (2012a). The geothermal heat flux is taken from Le Brocq et al. (2010). The basal sliding is similar to that of Schoof (2005), combining power-law and Coulomb behaviour. The grounding-line location is determined using hydrostatic equilibrium and sub-element parameterization (Gladstone et al., 2010; Leguy et al., 2014). The calving front is initialized from present-day observations and thereafter is allowed to retreat but not advance. See Lipscomb et al. (2019) for more information about the model.

### A4 FETI ULB: fast Elementary Thermomechanical Ice Sheet model (f.ETHISh v1.2)

The f.ETISh (fast Elementary Thermomechanical Ice Sheet) model (Pattyn, 2017) is a vertically integrated hybrid (SSA for basal sliding; SIA for grounded ice deformation) finite-difference ice sheet–ice shelf model with vertically integrated thermomechanical coupling. The transient englacial temperature field is calculated in a 3-D fashion. The marine boundary is represented by a grounding-line flux condition

according to Schoof (2007), coherent with power-law basal sliding (power-law coefficient of 2). Model initialization is based on an adapted iterative procedure based on Pollard and DeConto (2012a) to fit the model as close as possible to present-day observed thickness and flow field (Pattyn, 2017). The model is forced by present-day surface mass balance and temperature (van Wessem et al., 2014) based on the output of the regional atmospheric climate model RACMO2 for the period 1979–2011. The mass balance–elevation feedback is taken into account and a positive degree day (PDD) model for surface melt was employed. Isostatic adjustment was included using an elastic lithosphere–relaxed asthenosphere (ELRA) model. The PICO model (Reese et al., 2018b) was employed to calculate sub-shelf melt rates based on present-day observed ocean temperature and salinity (Schmidtko et al., 2014) on which the LARMIP forcings for the different basins are added. The model is run on a regular grid of 16 km with time steps of 0.1 years.

### A5   GRIS LSC: Grenoble Ice Sheet and Land Ice (GRISLI)

The GRISLI model is a three-dimensional thermomechanically coupled ice sheet model originating from the coupling of the inland ice model of Ritz (1992) and Ritz et al. (1997) and the ice shelf model of Rommelaere and Ritz (1996), extended to the case of ice streams treated as dragging ice shelves (Ritz et al., 2001). In the version used here, over the whole domain, the velocity field consists of the superposition of the shallow-ice approximation (SIA) velocities for ice flow due to vertical shearing and the shallow-shelf approximation (SSA) velocities used as a sliding law (Bueler and Brown, 2009). For the LARMIP experiments, we used the GRISLI version 2.0 (Quiquet et al., 2018), which includes the analytical formulation of Schoof (2007) to compute the flux at the grounding line. Basal drag is computed with a power-law basal friction (Weertman, 1957). For this study, we use an iterative inversion method to infer a spatially variable basal drag coefficient that ensures an ice thickness as close as possible to observations with a minimal model drift (Le clec'h et al., 2019a). The basal drag is assumed to be constant for the forward experiments. The model uses finite differences on a staggered Arakawa C grid in the horizontal plane at 16 km resolution with 21 vertical levels. Atmospheric forcing, namely near-surface air temperature and surface mass balance, is taken from the 1979–2014 climatological annual mean computed by the RACMO2.3 regional atmospheric model (van Wessem et al., 2014). Initial sub-shelf basal melting rates are the regionally averaged basal melting rates that ensure a minimal ice shelf thickness Eulerian derivative in a forward experiment with constant climate and a fixed grounding-line position. The initial ice sheet geometry, bedrock, and ice thickness are taken from the Bedmap2 dataset (Fretwell et al., 2013), and the geothermal heat flux is from Shapiro and Ritzwoller (2004).

### A6   IMAU UU: IMAUICE – IMAU/Utrecht University

The finite-difference model (de Boer et al., 2014) uses a combination of SIA and SSA solutions, with velocities added over grounded ice to model basal sliding (Bueler and Brown, 2009). The model grid at 32 km horizontal resolution covers the entire Antarctic ice sheet and surrounding ice shelves. The grounded ice margin is freely evolving, while the shelf extends to the grid margin and a calving front is not explicitly determined. We use the Schoof flux boundary condition (Schoof, 2007) at the grounding line with a heuristic rule following Pollard and DeConto (2012b). For the LARMIP experiments, the sea level equation is not solved or coupled (de Boer et al., 2014).

We run the thermodynamically coupled model with constant present-day boundary conditions to determine a thermodynamic steady state. The model is first initialized for 100 kyr using the average 1979–2014 surface mass balance (SMB) and surface ice temperature from RACMO2.3 (van Wessem et al., 2014). Bedrock elevation is fixed in time with data taken from the Bedmap2 dataset (Fretwell et al., 2013), and geothermal heat flux data are from Shapiro and Ritzwoller (2004). We then run for 30 kyr with constant ice temperature from the first run to get to a dynamic steady state, which is our initial condition. Model set-up, parameter settings, and initialization are identical to the IMAUICE submission to initMIP-Antarctica.

### A7   ISSM JPL: Ice Sheet System Model – JPL

The finite-element Ice Sheet System Model (Larour et al., 2012) is used with the two-dimensional shelfy-stream approximation (MacAyeal, 1989) over the entire Antarctic ice sheet. The model resolution varies between 1 km along the coast and 50 km in the interior of the domain, with the resolution of the ice shelves below 8 km. The model is initialized to match present-day conditions. On grounded ice, the viscosity is derived from a steady-state temperature that does not vary during the simulation, following Cuffey and Paterson (2010). The basal friction and the viscosity of floating ice are inferred to best match observed surface velocity (Rignot et al., 2011) using data assimilation (Morlighem et al., 2010). The basal sliding law follows a Budd friction law (Budd et al., 1979) that depends on the ice effective parameterization. The grounding-line position is determined using hydrostatic equilibrium, with sub-element parameterization of the friction (Seroussi et al., 2014). The melt rate is applied only for fully floating elements (Seroussi and Morlighem, 2018) and is initialized using mean rates of ocean estimates over the 2004–2015 period (Schodlok et al., 2016) that are kept constant with time. The surface mass balance is from the RACMO2.1 1979–2010 mean (Lenaerts et al., 2012). The ice front position is fixed at the extent of the present-day ice sheet. After initialization, the model is relaxed for 2 years so that the geometry and grounding lines can adjust (Seroussi et

al., 2011). For more details on the model and the initialization procedure, we refer to Schlegel et al. (2018), as we used a similar procedure here.

## A8    ISSM UCI: Ice Sheet System Model – UCI

We use the Ice Sheet System Model (ISSM; Larour et al., 2012) with a higher-order stress balance (Pattyn, 2003). The model resolution varies from 3 km around the coast to 50 km in the interior of the ice sheet, vertically extruded into 10 layers using a smaller spacing near the bed. The model is initialized using data assimilation of present-day conditions (Morlighem et al., 2013). We perform the inversion of basal friction assuming that the ice is in thermomechanical steady state based on a Budd friction law (Budd et al., 1979). The ice temperature is updated as the basal friction and internal deformation changes, and the ice viscosity is changed accordingly. At the end of the inversion, basal friction, ice temperature, and stresses are all consistent. After that, the model is run forward assuming that the temperature does not change. We use the surface mass balance from the RACMO2.1 1979–2010 mean (Lenaerts et al., 2012). The grounding line is parameterized using a sub-element friction scheme (Seroussi et al., 2014) and no melt in partially floating elements (Seroussi and Morlighem, 2018). The ice front is fixed through time. More details on the model are available in the ISMIP6 iniMIP-Antarctica study (Seroussi et al., 2019).

## A9    MALI LANL: model for prediction across scales – Albany Land Ice

MPAS-Albany Land Ice (MALI) (Hoffman et al., 2018) uses a three-dimensional, 1st-order Stokes approximation (Blatter–Pattyn) momentum balance solver using finite-element methods. Ice velocity is solved on a two-dimensional, map plane triangulation extruded vertically to form tetrahedra. Mass and tracer transport occur on the Voronoi dual mesh using a mass-conserving, finite-volume, 1st-order upwinding scheme. To ensure that the grounding line is captured by adequate spatial resolution even under full retreat of West Antarctica (or large parts of East Antarctica), mesh resolution is 2 km along grounding lines, in all marine regions of West Antarctica, and in marine regions of East Antarctica where present-day ice thickness is less than 2500 m. Mesh resolution coarsens to 20 km in the ice sheet interior and is no greater than 6 km within the large ice shelves. The horizontal mesh has 1.6 million cells. The mesh uses 10 vertical layers that are finest near the bed (4 % of total thickness) and coarsen towards the surface (23 % of total thickness). Ice temperature is based on results from Van Liefferinge and Pattyn (2013) and held fixed in time. The model uses a linear basal friction law with a spatially varying basal friction coefficient. The basal friction of grounded ice and the viscosity of floating ice are inferred to best match observed surface velocity (Rignot et al., 2011) using an adjoint-based optimization method (Perego et al., 2014) and then kept constant in time. The grounding-line position is determined using hydrostatic equilibrium, with a sub-element parameterization of the friction (analogous to SE3 from Seroussi et al., 2014). Sub-ice-shelf melt rates come from Rignot et al. (2013) and are extrapolated across the entire model domain to provide non-zero ice shelf melt rates after grounding-line retreat. The surface mass balance is the 1979–2010 mean from RACMO2.1 (Lenaerts et al., 2012). Maps of surface and basal mass balance forcing are kept constant with time. The ice shelf calving front positions are fixed at the extent of their present-day observations. To minimize large, non-physical transients resulting from the optimization procedure, the model is first relaxed by integrating forward in time for a century under steady forcing. During this time the model velocities, geometry, and grounding lines are free to adjust as needed.

## A10    PISM AWI: Parallel Ice Sheet Model – AWI

The Parallel Ice Sheet Model (Bueler and Brown, 2009; Winkelmann et al., 2011) in the hybrid shallow approximation is applied at 16 km resolution over the entire Antarctic ice sheet. The model is initialized via a 100 kyr equilibrium-type spin-up with steady present-day climate and fixed bedrock topography. The initial geometry is Bedmap2 (Fretwell et al., 2013). Basal friction is parameterized by the water content in the till and the depth of the ice base. Basal sliding is calculated via a pseudo-plastic friction law (Bueler and Brown, 2009; Winkelmann et al., 2011) depending on the yield strength of the till and the stored basal water. The grounding line is determined by hydrostatic equilibrium with a sub-grid parameterization of basal conditions (Feldmann et al., 2014). Both the grounding line and ice shelf front can freely evolve in the spin-up and the projections. Calving is governed by strain rate (eigencalving; Levermann et al., 2012) and ice shelf thickness (thickness calving). Calving is further applied if the ice extends over the continental shelf (sea floor below −2000 m). The melt rate underneath ice shelves is applied only to fully floating cells (no sub-grid basal melt) and calculated via the local difference between ocean temperature and pressure melting point. In the Amundsen and Bellingshausen Sea as well as underneath the Filchner Ice Shelf melt rates are modified by a scaling factor to better fit present-day patterns. Local ocean temperature is derived via extrapolation of 3-D ocean temperature fields from the World Ocean Atlas 2009 (Locarnini et al., 2013) for the present day. Present-day surface mass balance and ice surface temperature are from RACMO2.3 (van Wessem et al., 2014).

## A11 PISM DMI: Danish Meteorological Institute's Parallel Ice Sheet Model

The Parallel Ice Sheet Model (PISM version 0.7) utilizes a hybrid system (Bueler and Brown, 2009) combining the shallow-ice approximation (SIA) and shallow-shelf approximation (SSA) on an equidistant polar stereographic grid of 16 km. The basal resistance is described as plastic till for which the yield stress is given by a Mohr–Coulomb formula (Bueler and Brown, 2009; Schoof, 2006). Assuming an ocean temperature of $-1.7\,°C$ and constant melting factor ($F_{melt} = 0.001$) sub-shelf melting follows Eq. (7) in Martin et al. (2011) and occurs only for fully floating grid points, while the grounding-line position is determined on a sub-grid space (Feldmann et al., 2014). The calving parameterization incorporates three sub-schemes: at the ice shelf margin calving occurs when the thickness is less than 150 m; ice shelves that extend into the depth ocean disintegrate; the stress field evaluates the eigencalving parameterization with the proportionality constant of $5 \times 10^{17}$ (Levermann et al., 2012). Monthly atmospheric forcing deduced from sub-daily ERA-Interim reanalysis products (Berrisford et al., 2011; Dee et al., 2011) covers the period 1979–2012. Its 2 m air temperature determines the ice surface temperature, while the total precipitation is considered to be snow accumulation due to negligible surface melting in Antarctica. This forcing has been applied to match present-day conditions during spin-up, in which grounded ice margins, grounding lines, and calving fronts evolve freely.

## A12 PISM PIK: Potsdam Parallel Ice Sheet Model

The Parallel Ice Sheet Model (Winkelmann et al., 2011) (PISM, dev version c10a3a6e from 3 June 2018, based on v1.0, with added basal melt modifier, see documentation at https://pism-docs.org/wiki/doku.php, last access: 6 January 2020) uses a hybrid of the shallow-ice approximation (SIA) and the two-dimensional shelfy-stream approximation of the stress balance (SSA; Bueler and Brown, 2009; MacAyeal, 1989) over the entire Antarctic ice sheet. Here we use a plastic sliding law, which is independent of ice base sliding velocity. The model domain is discretized on a regular rectangular grid with 4 km horizontal resolution and a vertical resolution between 48 m at the top of the domain at 6000 and 7 m at the base of the ice. The model is initialized from Bedmap2 geometry (Fretwell et al., 2013) with model parameters (e.g. enhancement factors for SIA and SSA both equal 1 here) that minimize dynamic changes over 600 years of constant present-day climatic conditions (no equilibrium spin-up). PISM is a thermomechanically coupled (polythermal) model based on the Glen–Paterson–Budd–Lliboutry–Duval flow law (Aschwanden et al., 2012) such that the enthalpy can evolve freely for given boundary conditions. Basal meltwater is stored in the till. The Mohr–Coulomb criterion relates the yield stress by parameterizations of till material properties to the effective pressure on the saturated till (Bueler and van Pelt, 2015). The till friction angle is a shear strength parameter for the till material property and is optimized iteratively in the grounded region such that mismatch of equilibrium and modern surface elevation (8 km) is minimized (analogous to the friction coefficient in Pollard and DeConto, 2012a). The grounding-line position is determined using hydrostatic equilibrium, with sub-grid interpolation of the friction (Feldmann et al., 2014). The melt rate is not interpolated across the grounding line and is calculated with the Potsdam Ice-shelf Cavity mOdel (PICO; Reese et al., 2018b), which calculates melt patterns underneath the ice shelves for given ocean conditions; here this includes mean values over the observational period 1975–2012 (Schmidtko et al., 2014). The basin mean ocean temperature in the Amundsen region of $0.46\,°C$ has been corrected to a lower value of $-0.37\,°C$ as an average from the neighbouring Getz Ice Shelf basin, assuming that colder conditions were prevalent in the pre-industrial period. In the experiments basal melt offsets are added to the evolving PICO melt rate pattern, while basal melt is only for fully floating grid cells. The near-surface climate, surface mass balance, and ice surface temperature are from the RACMO2.3p2 1986–2005 mean (van Wessem et al., 2018) remapped from 27 km resolution. The calving front position can freely evolve using the eigencalving parameterization (Levermann et al., 2012) with $K = 1 \times 10^{17}$ m s and a terminal thickness threshold of 200 m.

## A13 PISM VUW: Parallel Ice Sheet Model – VUW

We use the Parallel Ice Sheet Model (PISM) version 0.7.1. PISM is a "hybrid" ice sheet–shelf model that combines shallow approximations of the flow equations that compute gravitational flow and flow by horizontal stretching (Bueler and Brown, 2009). The combined stress balance allows for a treatment of ice sheet flow that is consistent across non-sliding grounded ice to rapidly sliding grounded ice (ice streams) and floating ice (shelves). As with most continental-scale ice sheet models, we use flow enhancement factors for the shallow-ice and shallow-shelf components of the stress regime (3.5 and 0.5, respectively), which allow us to adjust creep and sliding velocities using simple coefficients. By doing so we are able to optimize simulations such that modelled behaviour is consistent with observed behaviour. The junction between grounded and floating ice is refined by a sub-grid-scale parameterization (Feldmann et al., 2014) that smooths the basal shear stress field and tracks an interpolated grounding-line position through time. This allows for much more realistic grounding-line motion, even with relatively coarse spatial grids, such as the 16 km grid used in our experiments. Surface mass balance is calculated using a positive degree day model that takes as inputs air temperature and precipitation from RACMO2.1 (Lenaerts et al., 2012). In previous simulations (e.g. Golledge et al., 2015) we have derived evolving melt beneath ice shelves from the thermody-

namic three-equation model of Hellmer and Olbers (1989), in which the melt rate is primarily controlled by salinity and temperature gradients across the ice–ocean interface. For the simplified experiments presented here, however, we set a spatially uniform melt rate as an initial condition and allow our modelled ice sheet to evolve in response to this. All of our simulations are initialized from a thermally and dynamically evolved state that represents the present-day ice sheet configuration and has a sea level equivalent volume of 58.35 m. We also run a control experiment, in which no additional basal melt is applied and which increases in volume by 0.05 m over 200 years.

### A14 PS3D PSU: Penn State University 3-D ice sheet model (PSUICE3D)

The model is described in detail in Pollard and DeConto (2012b), with updates in Pollard et al. (2015). The dynamics use a hybrid combination of vertically averaged SIA and SSA scaling. Floating ice shelves and grounding-line migration are included, with sub-grid interpolation for grounding-line position. The Schoof (2007) boundary layer formulation is imposed as a condition on ice velocity across the grounding line, which enables grounding-line migration to be simulated reasonably accurately without much higher grid resolution. The model includes standard equations for the evolution of ice thickness and internal ice temperatures with 10 unevenly spaced vertical layers. Bedrock deformation under the ice load is modelled as an elastic lithospheric plate above local isostatic relaxation (ELRA). Basal sliding follows a Weertman-type power law, occurring only where the bed is close to the melt point. Basal sliding coefficients are determined by an inverse method (Pollard and DeConto, 2012a), iteratively matching ice surface elevations to modern observations. Calving of ice shelves depends on combined depths of surface and basal crevasses relative to the ice shelf thickness. Crevasse depths depend primarily on the divergence of the ice velocity. The recently proposed mechanisms of hydrofracturing by surface meltwater and structural failure of large ice cliffs (DeConto and Pollard, 2016; Pollard et al., 2015) are not enabled for the LARMIP experiments. Oceanic melting at the base of ice shelves depends on the squared difference between nearby 400 m depth climatological ocean temperature (Levitus et al., 2012) and the melt point at the bottom of the ice. Atmospheric temperatures and precipitation are obtained from the ALBMAP climatology (Le Brocq et al., 2010), with an imposed sinusoidal cycle for monthly air temperatures. A simple box model based on positive degree days is used to compute annual surface mass balance, allowing for refreezing of meltwater. For the LARMIP experiments the model grid size is 16 km, and the control is spun up to equilibrium using perpetual modern climate forcing.

### A15 SICO ILTS: SICOPOLIS (SImulation COde for POLythermal Ice Sheets)

The model SICOPOLIS version 5.1 (http://www.sicopolis. net, last access: 6 January 2020) is applied to the Antarctic ice sheet with hybrid shallow-ice–shelfy-stream dynamics for grounded ice (Bernales et al., 2017) and shallow-shelf dynamics for floating ice. Ice thermodynamics are treated with the melting cold–temperate transition surface (CTS) enthalpy method (ENTM) by Greve and Blatter (2016). The ice surface is assumed to be traction-free. Basal sliding under grounded ice is described by a Weertman–Budd-type sliding law with sub-melt sliding (Sato and Greve, 2012) and subglacial hydrology (Calov et al., 2018; Kleiner and Humbert, 2014). The basal sliding coefficient is chosen differently for the 18 IMBIE 2016 basins (Rignot and Mouginot, 2016) to optimize the agreement between simulated and observed present-day surface velocities (Greve et al., 2019). The model is initialized to the reference year 1990 by a paleoclimatic spin-up over 140 000 years, forced by Vostok $\delta D$ converted to $\Delta T$ (Petit et al., 1999), in which the topography is nudged towards the present-day topography to enforce a good agreement. In the future climate simulations, the ice topography evolves freely. For the last 2000 years of the spin-up and all future climate simulations, a regular (structured) grid with 8 km resolution is used. In the vertical, we use terrain-following coordinates with 81 layers in the ice domain and 41 layers in the thermal lithosphere layer below. The present-day surface temperature is parameterized (Fortuin and Oerlemans, 1990), the present-day precipitation is by Arthern et al. (2006) and Le Brocq et al. (2010), and runoff is modelled by the positive degree day method with the parameters by Sato and Greve (2012). The 1960–1989 average SMB correction that results diagnostically from the nudging technique is used as a prescribed SMB correction for the future climate simulations. The bed topography is Bedmap2 (Fretwell et al., 2013), the geothermal heat flux is by Martos et al. (2017), and isostatic adjustment is included using an elastic lithosphere–relaxing asthenosphere (ELRA) model (parameters by Sato and Greve, 2012). Present-day ice shelf basal melting is parameterized by the ISMIP6 standard approach, a non-local quadratic melting parameterization that depends on the thermal forcing (ocean temperature minus freezing temperature) at the ice–ocean interface, and is tuned separately for the IMBIE 2016 basins (http://tinyurl.com/ismip6-wiki-ais). The LARMIP forcings (1, 2, 4, 8, 16, and 32 m yr$^{-1}$) for the five oceanic sectors are added to this parameterization.

### A16 ÚA UNN: University of Northumbria, Newcastle upon Tyne, UK

ÚA is a finite-element ice flow model (https://github.com/ GHilmarG/UaSource/, last access: 6 January 2020) that solves the momentum and mass conservation equations in

a vertically integrated form using the shallow-ice-stream approximation (SSA) (Gudmundsson et al., 2012). The transient evolution of the geometry is solved in a fully implicit manner, i.e. implicitly with respect to both velocities and ice thickness. The model uses automated mesh refinement and coarsening based on user-specified criteria. In the runs used in the study, mesh resolution ranged from about 1 to 40 km. The Weertman sliding law and Glen's flow law were used to describe basal sliding and ice rheology, respectively. Here the stress exponents of both laws were set to 3. Spatial variations in the sliding coefficient ($C$ in the Weertman sliding law) and rate factor ($A$ in Glen's flow law) were determined by conducting an inversion using the adjoint method with horizontal velocities as measurements using Tikhonov regularization on both amplitudes and second spatial derivatives. The ocean model MIT GCM (Massachusetts Institute of Technology general circulation model; http://mitgcm.org/, last access: 6 January 2020) has recently been coupled to ÚA (De Rydt et al., 2016). All runs presented were conducted by the co-author Jim Jordan.

**Code and data availability.** Data and analysis software can be obtained from the corresponding author upon request. The data can also be downloaded directly from http://www.pik-potsdam.de/~anders/larmip `TS11` and the analysis software from https://github.com/ALevermann/Larmip2019 `TS12`.

**Supplement.** The supplement related to this article is available online at: https://doi.org/10.5194/esd-11-1-2020-supplement.

**Author contributions.** AL designed and coordinated the study and computed the projections. All other authors contributed their model simulations as well as to the writing of the paper and the discussion of the results.

**Competing interests.** The authors declare that they have no conflict of interest.

**Acknowledgements.** We would like to thank two anonymous reviewers and Daniel Gilford for extremely helpful comments on the paper.

Support for Daniel Martin, Tong Zhang, Matthew J. Hoffman, Mauro Perego, Stephen F. Price, and Esmond Ng was provided through the Scientific Discovery through Advanced Computing (SciDAC) programme funded by the US Department of Energy (DOE), Office of Science, Biological and Environmental Research, and Advanced Scientific Computing Research programmes. Their contributions relied on computing resources from the National Energy Research Scientific Computing Center, a DOE Office of Science user facility supported by the Office of Science of the US Department of Energy under contract no. DE-AC02-05CH11231.

Christian Rodehacke has received funding from the European Research Council under the European Community's Seventh Framework Programme (FP7/2007–2013)/ERC grant agreement 610055 as part of the Ice2Ice project.

Heiko Goelzer has received funding from the programme of the Netherlands Earth System Science Centre (NESSC), financially supported by the Dutch Ministry of Education, Culture and Science (OCW) under grant no. 024.002.001.

A portion of this research was carried out at the Jet Propulsion Laboratory, California Institute of Technology, under a contract with the National Aeronautics and Space Administration. Helene Seroussi and Nicole-Jeanne Schlegel were supported by grants from the NASA Cryospheric Science, Sea Level Change Team, and Modeling Analysis and Prediction programmes.

Ralf Greve was supported by the Japan Society for the Promotion of Science (JSPS) under KAKENHI grant nos. JP16H02224, JP17H06104, and JP17H06323.

The work of Thomas Kleiner and Angelika Humbert has been conducted in the framework of the PalMod project (FKZ: 01LP1511B), supported by the German Federal Ministry of Education and Research (BMBF) as a Research for Sustainability initiative (FONA).

The material provided for the CISM model is based upon work supported by the National Center for Atmospheric Research, which is a major facility sponsored by the National Science Foundation under cooperative agreement no. 1852977. Computing and data storage resources, including the Cheyenne supercomputer (https://doi.org/10.5065/D6RX99HX `TS13`), were provided by the Computational and Information Systems Laboratory (CISL) at NCAR.

Torsten Albrecht was supported by the Deutsche Forschungsgemeinschaft (DFG) in the framework of the priority programme "Antarctic Research with comparative investigations in Arctic ice areas" by grants LE1448/6-1 and LE1448/7-1. Julius Garbe acknowledges funding from the Leibniz Association (project DominoES).

Jonas Van Breedam and Philippe Huybrechts acknowledge support from the iceMOD project funded by the Research Foundation – Flanders (FWO-Vlaanderen).

Malte Meinshausen received funding from the National Science Foundation (NSF grant no. 1739031) through the PROPHET project, a component of the International Thwaites Glacier Collaboration (ITGC).

**Financial support.** This research has been supported by the U.S. Department of Energy (grant no. DE-AC02-05CH11231), the European Research Council (ICE2ICE (grant no. 610055)), the Dutch Ministry of Education, Culture and Science (grant no. 024.002.001), the Japan Society for the Promotion of Science (grant nos. JP16H02224, JP17H06104, and JP17H06323), the German Federal Ministry of Education and Research (BMBF) FONA (grant no. 01LP1511B), the German Research Foundation (grant nos. LE1448/6-1 and LE1448/7-1), and the National Science Foundation (grant no. 1852977).

The article processing charges for this open-access publication were covered by the Potsdam Institute for Climate Impact Research (PIK).

**Review statement.** This paper was edited by Yun Liu and reviewed by Daniel Gilford and two anonymous referees.

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

## Remarks from the language copy-editor

## Remarks from the typesetter