# Peer review of "Projecting Antarctica's contribution to future sea level rise from basal ice-shelf melt using linear response functions of 16 ice sheet models (LARMIP-2)"

_Earth System Dynamics, 2019_

## Referee Comment (RC1) · Anonymous Referee #1 · 3 Jul 2019

In this manuscript, the authors use the linear impulse-response model previously presented in Levermann et al. (2014) to fit Antarctic ice sheet simulations from sixteen different ice sheet models, allowing the construction of a simple statistical emulator of these models' behavior in five aggregate ice-sheet sectors. This is a useful tool for producing future probabilistic sea-level projections.

It is, however, somewhat regrettable that the paper does not give more attention to at least attempting to explain the differences among ice-sheet responses – the differences among models that can be observed in figures 4 and 5 are stark, and some

discussion would be very welcome. As just one example, there are major differences among models in the amount of high-frequency variability, and an attempt to put some explanatory taxonomy on these differences would be extremely useful.

What are we to take away from the GCMs with poor fits and long delays (e.g., IPSL-CM5A-MR and MRI-CGCM3)? Is it valid to include these in projecting the forcing? It would be helpful to have supplemental figures showing the underlying data.

It would also be useful to have some quantitative measure of the quality of the linear response approximation for each ice sheet model. This can be eyeballed from Figure 4, but it'd be better to have an objective metric.

Structurally and stylistically, the paper has some significant flaws that hinder its comprehensibility. Most importantly, the paper clearly expects the reader to have recently read Levermann et al 2014. While I understand the authors' desire not to be repetitive, the paper needs to be comprehensible on its own. A clear summary of the assumptions and approach of Levermann et al 2014 needs to be provided early on. The second half of section 2.4, which provides the equations underlying the statistical emulator, comes way too late – the structure of the emulator needs to be clearly described first, followed by a clear description of the calibration, and then followed by a clear description of the projection method. The current version is all tangled together.

In the discussion, I would encourage the authors to contextualize their results based on other available information. For example, the structured expert judgement study of Bamber et al. (2019, 10.1073/pnas.1817205116) might provide useful context.

The authors' tone is also peculiarly (and conversationally) beseeching and apologetic. It is important for the authors to be clear about the limitations of the approach, but polite imperatives ("Please note this", "Please note that") are excessively circuitous; the authors should just state what they wish the reader to note (and do it in a place that makes logical sense, given the flow of the paper).

[Figure]

On page 10, the last paragraph seems a bit excessive given the decision not to weight the different models. I would either drop the equation and keep only a qualitative discussion of model weighting, or (perhaps better) do a sensitivity study using model weighting.

The paper is also missing a proper conclusion. A paragraph explaining the decision not to weight is a peculiar way to conclude.

Figure 1: "intervall" -> "interval"

Tables 11 and 12 have no units.

---

## Short Comment (SC1) · 4 Jul 2019

Dear reviewer,

thank you for your review! We will definitely provide the additional analysis, conclusions and literature embedding. The additional work that is required is certain to improve the manuscript and we will definitely do it.

Best wishes, Anders

---

## Short Comment (SC2) · 10 Jul 2019

This study uses a linear response function method to estimate the basal melting of 16 ice sheet models in response to a range of external forcing from anthropogenic emissions. The study is relatively novel using a recently published methodology (Levermann et al. 2014) to explore basal melt across the many state-of-the-art ice sheet models, and is a natural follow up on that work. The authors succinctly describe their methods and results, and promisingly find relatively good agreement between models and with observations. I have only a few substantive comments which I hope the authors will

consider. They relate to the motivational and concluding discussions, and the need for more detailed statistical analyses of the linear assumption; my other comments concern methodological clarity or are grammatical. In my estimation this study will be a valuable addition to the literature.

General Comments:

1. The abstract and conclusions provide new information to the reader that are not discussed throughout the rest of the paper, which detracts from the study. In the case of the abstract, the Paris agreement is mentioned without being discussed in the main text, and the method's structural uncertainty limitation isn't discussed anywhere except in the abstract. It would be helpful discuss these somewhere in the paper itself, or removed from the abstract. It is also confusing that a new methodology is specifically introduced in the discussion section (pg. 10, line 15). I would suggest introducing it earlier in the paper (section 3) where the model is compared against observational trends, and then simply noting the method (without the equation) in the concluding section.

2. One of the more compelling sentences in the abstract discussed how this study will be used to explore the range of responses to the external climate system forcing. But that motivation/wording isn't found in the introduction, which is a missed opportunity. The paper would be improved if a paragraph which more strongly motivated this work was written, becoming the second paragraph of section 1.

3. Section 2.3 and pg. 5, lines 22-34: While the criteria for choosing 8 m/yr was explained here, the way this section is currently written doesn't provide a clear explanation of what the 8 m/yr value is trying to accomplish and why a specific melt rate was selected in the first place. This makes the method sections on the response function difficult to follow. A few sentences are needed to clarify and describe how the 8 m/yr value is used to derive the response function, and then connected to how those functions are used along with the range of melting rates in Figure 3 to derive sea level

contributions. It will be helpful to methodically explain what the "switch-on" experiments are, and how they relate to the timeseries of melt rates found.

4. In pg. 7, lines 11-14: Is there any way to discuss and quantify how close "generally close" is, maybe with some statistical test? It seems that as alpha approaches 0 or 1, you are off by a factor of 2 or more, which probably does not constitute linearity. How was the range -1 to 2 chosen? A theoretically supported statistical test for linearity would be helpful for making sense of the results.

5. Section 3 and Figures 6-7: How do these observations compare with the modeled changes over individual regions?

6. A concluding paragraph noting the importance and novelty of this work—for understanding the uncertainties in ice-sheet response to uncertainties in external forcing—would improve the manuscript. Moreover, the importance of human decisions (i.e. their role in driving external forcing and its uncertainties) should be emphasized.

Line-by-line comments:

Pg. 2, Line 4: This sentence reads a bit odd, can it be rewritten for clarity?

Lines 5-6: This sentence might be better suited towards the end of the abstract, or just in main text of the paper. Noting the limitations here in the abstract disrupts the flow of what it is trying to highlight and the study goals. In contrast, the limitations discussed in the following sentences seem more pertinent to the study goals.

Line 23: To my knowledge, the total contributions from the Paris Agreement could put the world on track to ∼3 degrees of warming or more, although ratcheting down emissions and could reduce this. In contrast the stated goal of the Paris agreement is "well below 2 degrees". I suggest either adding this language of "goal", or removing this from the abstract. Moreover, the Paris agreement is not mentioned anywhere else in the manuscript, so it may be appropriate to either cut it altogether or add more details to the main text.

[Figure]

Line 26: Remove "the" before "five Antarctic regions". The reader doesn't yet have familiarity with which 5 regions are being referred to; this is good opportunity to let them know that you are using five regions.

Line 26-27: "rate" should be "rates", "is" should be "are"

Pg. 3, line 17: remove "a"

Line 18: "will be" should be "is"

Line 20: I suggest removing the line about repeating the method. This comment could be left to section 2, and "try not" reads rather casual and should be rewritten.

Line 29: I suggest removing this casual line about the contribution the paper is "trying to make".

Line 35: "This will be pointed out. . ." should be removed.

Line 36: This sentence is helpful! A useful detail which sets up the next section nicely.

Lines 38 and 39: "precisely" probably isn't necessary here.

Line 40: This sentence about "the only thing that changed" is casual and not very descriptive. I suggest rewriting as: "The only difference between our study and the previous one are the ice sheet models used to project ice-sheet changes and sea level rise contributions."

Pg. 4, line 1: Please replace "carbon dioxide concentration" with something like "emission pathways". My understanding is that RCP scenarios are in carbon dioxide equivalent, including not only CO2 but also other GHGs and constituents (and their forcings), which differ not only in concentrations but substance between each pathway.

Line 4: remove "the"

Line 10: Are the samples in this bulk approach approximately equally distributed among all ice-sheet models? This could affect the results of Figure 7, for instance, because

the weighting towards some models might bias the results (cf. Figure 6 where there are clear differences between different models).

Line 13: "different". . . is each timeseries drawn "without replacement"? Are any random selections repeated in your 20,000 samples?

Line 23: replace "now given" with "given below".

Line 25: This introduction to RCPs is probnably unnecessary, as they were introduced above in lines 1-2.

Pg. 5, lines 33-34: This sentence reads very awkwardly. I recommend rewriting this sentence for clarity as, "This is the most balanced choice to span the range of simulations, with 4 m/yr being too low for most of the RCP-8.5 scenario and 16 m/yr being too high for the majority of scenarios and ensemble samples."

Line 37: can you add "(described in appendix A)" after "ice sheet models."?

Line 37 and section 2.5: This paragraph seems out of place as its own section, being relatively short and quite necessary for the discussion at the beginning of section 2.4 I recommend removing section 2.5 altogether, and moving its context/text to wrap into and support the first few sentences of section 2.4.

Line 37: There are switches between past and present tense throughout. Can you choose one and be consistent in the manuscript?

Pg. 6, line 3: The start of this sentence could be rewritten for clarity as, "Although these simulations are highly interesting, a full discussion of their results..."

Line 6: remove the redundant "the uncertainty in."

Line 22: please add ", A_mu(t)," after "the observed response" to note the equation terms.

Line 24: This section is a bit difficult to follow. I suggest removing "the response func-
tion is obtained" and adding a sentence like, "Following this procedure and using the fixed Heavyside forcing mu=8m/yr chosen above, we obtain the response functions for each of the ice-sheet models."

Line 38: I suggest rewriting "for the 200 years of the forcing period" as "which is held constant over 200 years."

Pg. 7, line 25-26: "but not that far off in most cases" is rather general. A short but descriptive finish to this sentence would improve and contextualize this paragraph. I suggest rewriting as: "but in most cases the assumption is a reasonable approximation of how basal melting is responding to external forcing."

Pg. 8, line 7: "and" should be "with"

Line 24: Increases in the ice-sheet associated with precipitation changes cannot be explicitly or implicitly accounted for in your linear response (this is apparent in the figures, as no changes are below 0). But observed ice-sheet changes over this period in the EAIS are increasing in mass. If they are not accounted for, could that also lead to differences between the estimates here and the observations? You mention this in the conclusions, but can you discuss this drawback and its implications in more detail in this section?

Line 31: By "largest" do you refer to the median values?

Lines 35-36: This sentence is difficult to follow.

Line 38: How are uncertainties distributed across the different regions, and why?

Pg. 9, line 1: Is there a way to decompose equation 5 at each timestep to determine the relative importance of each component at each timestep?

Line 6: "However" is redundant with your previous statement and I do not think is what you mean. Can you use something like "In any case..."

Pg. 11: Are the codes and data used to produce these analyses publicly available? Please provide to the extent possible, in accordance with the ESD data policy: https://www.earth-system-dynamics.net/about/data_policy.html

Figure 1 caption: It would be helpful to add the bold symbols for the basal melt rate and sensitivity, as was done for the other terms.

Figure 4b-e captions: "Figure 3a" should be "Figure 4a".

Figure 5a: Please note that the response function are the grey lines.

Tables 2-5: Are these coefficients the alpha_r values? It would be helpful label them in the table with their symbol and describe them in words in the caption.

---

## Short Comment (SC3) · 10 Jul 2019

Dear Daniel,

thanks for the extensive review. We will definitely address your points during the revisions.

Bests, Anders
* * *
[Figure]

2019.

---

## Referee Comment (RC2) · Anonymous Referee #2 · 12 Jul 2019

The study uses a linear response function method earlier used by (Levermann et al. 2014) to estimate the basal melting of ice sheet models in response to a range of external forcing from anthropogenic emissions. Instead of 5 ice sheet models used in the earlier study, it has been extended to 16 Ice sheet models.

Though the methodology was adopted earlier, it would be interesting to understand the response across the many state-of-the-art ice sheet models.

Below are the general and specific comments.

[Figure]

General Comments:

1. The authors expect the reader to have a clear understanding of Levermann et al. 2014. A summary of the work would be excellent.

2. Some details in the abstract are not referenced in the text, e.g., the Paris Agreement (citation and a brief description would be excellent)

3. Section 2.6 Validity of Linearity assumption." The authors fail to clarify upon what they mean by "The alpha values are generally close to zero, which represent linearity."

Line-by-line comments:

Page 2 Line 19: "For the so-called business-as-unusual.." - please rephrase

Page 2 Line 23: "Paris Climate Agreement" - Missing citation. Do not see any reference related to this later in the text.

Page 3 Line 7: Clausisus-Clapeyron law - missing citation

Page 3 Line 28-29: "The advantage here is that we can investigate the response of the models to the full range of uncertain forcing and combine this for all the different ice sheet models. That is the main contribution this study is trying to make" - Please combine the sentences.

Page 3 Line 30: In addition - missing comma

Page 3 Line 33: "It is important to note that in this study" - missing comma

Page 3 Line 35: "In any case whenever the term Antarctic contribution to sea-level rise is used this refers to the sea-level relevant ice loss induced from basal ice shelf melting only." - please rephrase may be "In this study, .."

Page 3 Line 40: "The only thing that changed is the ice sheet models." - Unclear sentence

Page 4 Line 21-23: "Although there are other possibilities, this approach preserves

the forcing structure as provided by the ocean models which is why we selected it." - please rephrase

Page 5 Line 37-38: Does it mean that the configuration was set up for each region illustrated in Figure 2. rather than the whole region of the Antarctic and presenting results for various sectors? If for each region, then what about the influence of boundaries between the regions?

Page 6 Line 2: "A number of modeling groups ..." - please include citations, a brief description about the modeling groups

Page 6 Line 3: "... beyond the scope" - it would be nice to have a brief description with citations.

Page 6 Line 8: "That however might sound worse than it is." - please rephrase

Page 7 Line 11: "The alpha values are, however, generally close to zero, which represents linearity" - Any references to substantiate the assumption or what does the author mean by saying "generally close to zero."

Page 7 Line 21: "While some models show an instantaneous ice loss response, most models exhibit a more gradual increase of the ice loss over time." - Please include a bit more detail, which is which.

Page 8 Line 3 "are started" - were started

Page 8 Line 10-14: Isn't it better use the model estimates that fall within the uncertainty range of observation to derive conclusions? Can you substantiate the reason for using all the models? It would be nice to see the total from the models whose estimates fall within the uncertainty range of observation.

Page 8 Lin 31: "Overall" insert a comma

Page 9 Line 39: "... compared to what might occur in reality." It would be nice to include details on some other studies for comparison.

---

## Short Comment (SC4) · 12 Jul 2019

Dear reviewer,

Thank you for taking on the review and for your comments. We will be addressing them in our revisions.

Best wishes, Anders
* * *
[Figure]

2019.

---

## Author Comment (AC1) · 14 Oct 2019

**Response to reviewers**

*We would like to thank the reviewers for the very constructive criticism of our manuscript and are glad that they consider the study in principle worthy for publication in ESD. We were able to address all requests including those of an extensive additional review by Daniel Gilford. These reviews have*
5 *considerably improved the manuscript for which we are grateful.*

*In particular, we have added a comparison of the projections to existing projections of the sea level contribution for Antarctica (as requested by both reviewer's and Daniel Gilford), a more extensive discussion of the results and methods and a more explicit description of the procedure used for projections.*

10 *In order to facilitate the reading of this document, responses to the reviewers are given in blue and italic compared to the reviewer comments which are given in* black without italic font.

**Response to reviewer #1**

In this manuscript, the authors use the linear impulse-response model previously presented in
15 Levermann et al. (2014) to fit Antarctic ice sheet simulations from sixteen different ice sheet models, allowing the construction of a simple statistical emulator of these models' behavior in five aggregate ice-sheet sectors. This is a useful tool for producing future probabilistic sea-level projections.

*Response: We are happy that the author sees value in the overall approach.*

20 It is, however, somewhat regrettable that the paper does not give more attention to at least attempting to explain the differences among ice-sheet responses – the differences among models that can be observed in figures 4 and 5 are stark, and some discussion would be very welcome. As just one example, there are major differences among models in the amount of high-frequency variability, and an attempt to put some explanatory taxonomy on these differences would be extremely useful.

25 What are we to take away from the GCMs with poor fits and long delays (e.g., IPSLCM5A-MR and MRI-CGCM3)? Is it valid to include these in projecting the forcing? It would be helpful to have supplemental figures showing the underlying data. It would also be useful to have some quantitative measure of the quality of the linear response approximation for each ice sheet model. This can be eyeballed from Figure 4, but it'd be better to have an objective metric.

30 *Response: We agree with the reviewer that a discussion of the differences between the models is highly desirable. We have added a discussion of the relation of each model to the overall performance of the models together with a possible explanation to section 4. A really deep analysis would be most desirable but is also incredibly hard to do for 16 models at once. We hope that the reviewer finds the discussion somewhat useful and agrees that the comparison of the results in the figures and tables bears*
35 *some value for the reader on its own.*

Structurally and stylistically, the paper has some significant flaws that hinder its comprehensibility.

Most importantly, the paper clearly expects the reader to have recently read Levermann et al 2014. While I understand the authors' desire not to be repetitive, the paper needs to be comprehensible on its
40 own. A clear summary of the assumptions and approach of Levermann et al 2014 needs to be provided early on. The second half of section 2.4, which provides the equations underlying the statistical

emulator, comes way too late – the structure of the emulator needs to be clearly described first, followed by a clear description of the calibration, and then followed by a clear description of the projection method. The current version is all tangled together.

*Response: We would like to thank the reviewer for highlighting this omission. We thought it would be best not to be too repetitive, but have now expanded the explanation in the method section 2.*

In the discussion, I would encourage the authors to contextualize their results based on other available information. For example, the structured expert judgement study of Bamber et al. (2019, 10.1073/pnas.1817205116) might provide useful context.

*Response: We fully agree and have added a discussion of the existing sea level and ice sheet projections literature for context.*

The authors' tone is also peculiarly (and conversationally) beseeching and apologetic. It is important for the authors to be clear about the limitations of the approach, but polite imperatives ("Please note this", "Please note that") are excessively circuitous; the authors should just state what they wish the reader to note (and do it in a place that makes logical sense, given the flow of the paper).

*Response: As the style of the writing is a matter of personal taste, I would appreciate if that was not part of the review process since it bears no scientific content. In order to oblige I have however eliminated all "pleases" from the text.*

On page 10, the last paragraph seems a bit excessive given the decision not to weight the different models. I would either drop the equation and keep only a qualitative discussion of model weighting, or (perhaps better) do a sensitivity study using model weighting.

*Response: We agree and have reduced the paragraph and shifted it into the comparison of the hindcast with observations, in order to provide some discussion for readers that wonder why there is no weighting.*

The paper is also missing a proper conclusion. A paragraph explaining the decision not to weight is a peculiar way to conclude.

*Response: We fully agree with the reviewer and have added an overall conclusion together with a discussion of the existing sea level projection literature.*

Figure 1: "intervall" -> "interval"

*Response: Done.*

Tables 11 and 12 have no units.

*Response: The units were given in the caption. Upon production we will conform to the journals convention on this.*

**Response to reviewer #2**

The study uses a linear response function method earlier used by (Levermann et al. 2014) to estimate the basal melting of ice sheet models in response to a range of external forcing from anthropogenic emissions. Instead of 5 ice sheet models used in the earlier study, it has been extended to 16 Ice sheet models. Though the methodology was adopted earlier, it would be interesting to understand the response across the many state-of-the-art ice sheet models.

Below are the general and specific comments.

General Comments:

1. The authors expect the reader to have a clear understanding of Levermann et al. 2014. A summary of the work would be excellent.

*Response: We fully agree and provide more explanation of the methodology used in Levermann et al. 2014 and have added a discussion and comparison of the results in the previous paper to the discussion and conclusion section.*

2. Some details in the abstract are not referenced in the text, e.g., the Paris Agreement (citation and a brief description would be excellent)

*Response: We have tried to fix this omission and hope that there are no more details in the abstract that are not referenced in the text anymore. In particular the Paris Agreement is now referenced in the introduction.*

3. Section 2.6 Validity of Linearity assumption." The authors fail to clarify upon what they mean by "The alpha values are generally close to zero, which represent linearity."

*Response: Thanks for pointing this out. We have added a longer explanation.*

Line-by-line comments:

Page 2 Line 19: "For the so-called business-as-unusual.." - please rephrase

*Response: Rephrased into "For the unabated warming path, RCP-8.5,…"*

Page 2 Line 23: "Paris Climate Agreement" - Missing citation. Do not see any reference related to this later in the text.

*Response: Done.*

Page 3 Line 7: Clausisus-Clapeyron law - missing citation

*Response: References were added.*

Page 3 Line 28-29: "The advantage here is that we can investigate the response of the models to the full range of uncertain forcing and combine this for all the different ice sheet models. That is the main contribution this study is trying to make" – Please combine the sentences.

*Response: Done.*

Page 3 Line 30: In addition - missing comma

*Response: Added in all places where it was missing.*

Page 3 Line 33: "It is important to note that in this study" - missing comma

*Response: I am not sure a comma is missing. I will wait for the proof readers if that is alright?*

Page 3 Line 35: "In any case whenever the term Antarctic contribution to sea-level rise is used this refers to the sea-level relevant ice loss induced from basal ice shelf melting only." - please rephrase may be "In this study, .."

*Response: Done.*

Page 3 Line 40: "The only thing that changed is the ice sheet models." – Unclear sentence

*Response: Done.*

Page 4 Line 21-23: "Although there are other possibilities, this approach preserves the forcing structure as provided by the ocean models which is why we selected it." - please rephrase

*Response: Done.*

Page 5 Line 37-38: Does it mean that the configuration was set up for each region illustrated in Figure 2. rather than the whole region of the Antarctic and presenting results for various sectors? If for each region, then what about the influence of boundaries between the regions?

*Response: We have added a discussion on this.*

Page 6 Line 2: "A number of modeling groups ..." - please include citations, a brief description about the modeling groups; Page 6 Line 3: "... beyond the scope" - it would be nice to have a brief description with citations.

*Response: In order to make this feasible we decided to provide all data of the simulations as supplementary information to the manuscript. Description in the paper would add a number of additional figures which would increase the number of figures even more. I hope the reviewer can agree to this procedure.*

Page 6 Line 8: "That however might sound worse than it is." - please rephrase

*Response: Done.*

Page 7 Line 11: "The alpha values are, however, generally close to zero, which represents linearity" - Any references to substantiate the assumption or what does the author mean by saying "generally close to zero."

*Response: We have omitted this statement and replaced it by an assessment of the linearity assumption. This might be better since the alpha values are given in the figures. It is thus easiest for the reader to directly compare the alpha values with the corresponding deviations of the linearity assumption from the model simulations. Thanks for the note.*

Page 7 Line 21: "While some models show an instantaneous ice loss response, most models exhibit a more gradual increase of the ice loss over time." - Please include a bit more detail, which is which.

*Response: We have added examples.*

Page 8 Line 3 "are started" - were started

*Response: Done.*

Page 8 Line 10-14: Isn't it better use the model estimates that fall within the uncertainty range of observation to derive conclusions? Can you substantiate the reason for using all the models? It would be nice to see the total from the models whose estimates fall within the uncertainty range of observation.

*Response: We have added an explanation both in this section and also in the conclusion as of why modelling the observed sea level contribution from Antarctica is not a good indicated for a successful projection of the future.*

Page 8 Lin 31: "Overall" insert a comma

*Response: Done.*

Page 9 Line 39: "... compared to what might occur in reality." It would be nice to include details on some other studies for comparison.

*Response: We have added a longer discussion of the models and also of other studies.*

**Response to Daniel Gilford's comments**

This study uses a linear response function method to estimate the basal melting of 16 ice sheet models in response to a range of external forcing from anthropogenic emissions. The study is relatively novel using a recently published methodology (Levermann et al. 2014) to explore basal melt across the many
5   state-of-the-art ice sheet models, and is a natural follow up on that work. The authors succinctly describe their methods and results, and promisingly find relatively good agreement between models and with observations. I have only a few substantive comments which I hope the authors will consider. They relate to the motivational and concluding discussions, and the need for more detailed statistical analyses of the linear assumption; my other comments concern methodological clarity or are grammatical. In my
10  estimation this study will be a valuable addition to the literature.

*Response: We are happy about the overall approval of the methodology and results.*

General Comments:

1. The abstract and conclusions provide new information to the reader that are not discussed throughout the rest of the paper, which detracts from the study. In the case of the abstract, the Paris agreement is
15  mentioned without being discussed in the main text, and the method's structural uncertainty limitation isn't discussed anywhere except in the abstract. It would be helpful discuss these somewhere in the paper itself, or removed from the abstract. It is also confusing that a new methodology is specifically introduced in the discussion section (pg. 10, line 15). I would suggest introducing it earlier in the paper (section 3) where the model is compared against observational trends, and then simply noting the
20  method (without the equation) in the concluding section.

*Response: We have added the uncertainty discussion of the abstract in the discussion section and the Paris Climate Agreement in the introduction. Further we have moved the methodology of the weighting procedure to section 3. These are important suggestions by the reviewer which have improved the manuscript.*

25  2. One of the more compelling sentences in the abstract discussed how this study will be used to explore the range of responses to the external climate system forcing. But that motivation/wording isn't found in the introduction, which is a missed opportunity. The paper would be improved if a paragraph which more strongly motivated this work was written, becoming the second paragraph of section 1.

*Response: We have added such a paragraph in the introduction.*

30  3. Section 2.3 and pg. 5, lines 22-34: While the criteria for choosing 8 m/yr was explained here, the way this section is currently written doesn't provide a clear explanation of what the 8 m/yr value is trying to accomplish and why a specific melt rate was selected in the first place. This makes the method sections on the response function difficult to follow. A few sentences are needed to clarify and describe how the 8 m/yr value is used to derive the response function, and then connected to how those functions are used
35  along with the range of melting rates in Figure 3 to derive sea level contributions. It will be helpful to methodically explain what the "switch-on" experiments are, and how they relate to the timeseries of melt rates found.

*Response: We have added an explanation to the end of the section. Thanks for the hint.*

4. In pg. 7, lines 11-14: Is there any way to discuss and quantify how close "generally close" is, maybe
40  with some statistical test? It seems that as alpha approaches 0 or 1, you are off by a factor of 2 or more, which probably does not constitute linearity. How was the range -1 to 2 chosen? A theoretically supported statistical test for linearity would be helpful for making sense of the results.

*Response: We have changed this paragraph in response to reviewer #2's comment. It was not well put, we agree.*

5. Section 3 and Figures 6-7: How do these observations compare with the modelled changes over individual regions?

*Response: This has to be part of another study, I am afraid. We will provide the data for later studies.*

6. A concluding paragraph noting the importance and novelty of this work—for understanding the uncertainties in ice-sheet response to uncertainties in external forcing—would improve the manuscript. Moreover, the importance of human decisions (i.e. their role in driving external forcing and its uncertainties) should be emphasized.

*Response: We have added a conclusion to the end of the paper.*

Line-by-line comments:

Pg. 2, Line 4: This sentence reads a bit odd, can it be rewritten for clarity?

*Response: Done.*

Lines 5-6: This sentence might be better suited towards the end of the abstract, or just in main text of the paper. Noting the limitations here in the abstract disrupts the flow of what it is trying to highlight and the study goals. In contrast, the limitations discussed in the following sentences seem more pertinent to the study goals.

*Response: We would like to keep the sentence there.*

Line 23: To my knowledge, the total contributions from the Paris Agreement could put the world on track to 3 degrees of warming or more, although ratcheting down emissions and could reduce this. In contrast the stated goal of the Paris agreement is "well below 2 degrees". I suggest either adding this language of "goal", or removing this from the abstract. Moreover, the Paris agreement is not mentioned anywhere else in the manuscript, so it may be appropriate to either cut it altogether or add more details to the main text. Line 26: Remove "the" before "five Antarctic regions". The reader doesn't yet have familiarity with which 5 regions are being referred to; this is good opportunity to let them know that you are using five regions.

*Response: We have added a reference and a reference to the discussion of the agreement.*

Line 26-27: "rate" should be "rates", "is" should be "are"

*Response: We would like to keep it as it is.*

Pg. 3, line 17: remove "a"

*Response: We would like to keep it as it is.*

Line 18: "will be" should be "is"

*Response: Done.*

Line 20: I suggest removing the line about repeating the method. This comment could be left to section 2, and "try not" reads rather casual and should be rewritten.

*Response: Done.*

Line 29: I suggest removing this casual line about the contribution the paper is "trying to make".

*Response: Done.*

Line 35: "This will be pointed out: : :" should be removed.

*Response: Done.*

Line 36: This sentence is helpful! A useful detail which sets up the next section nicely.

*Response: Thanks.*

Lines 38 and 39: "precisely" probably isn't necessary here.

*Response: Removed.*

Line 40: This sentence about "the only thing that changed" is casual and not very descriptive. I suggest rewriting as: "The only difference between our study and the previous one are the ice sheet models used to project ice-sheet changes and sea level rise contributions."

*Response: Done.*

Pg. 4, line 1: Please replace "carbon dioxide concentration" with something like "emission pathways". My understanding is that RCP scenarios are in carbon dioxide equivalent, including not only CO2 but also other GHGs and constituents (and their forcings), which differ not only in concentrations but substance between each pathway.

*Response: We know and understand but it would be rather tedious to go into this here. Especially since RCPs are NOT emission pathways which would make it much less precise.*

Line 4: remove "the"

*Response: We would like to keep it as it is.*

Line 10: Are the samples in this bulk approach approximately equally distributed among all ice-sheet models? This could affect the results of Figure 7, for instance, because the weighting towards some models might bias the results (cf. Figure 6 where there are clear differences between different models).

*Response: No weighting was applied.*

Line 13: "different": : : is each timeseries drawn "without replacement"? Are any random selections repeated in your 20,000 samples?

*Response: In each step the time series is newly selected from the same ensemble. We believe that should be clear here.*

Line 23: replace "now given" with "given below".

*Response: Done.*

Line 25: This introduction to RCPs is probnably unnecessary, as they were introduced above in lines 1-2.

*Response: We would like to keep it as it is.*

Pg. 5, lines 33-34: This sentence reads very awkwardly. I recommend rewriting this sentence for clarity as, "This is the most balanced choice to span the range of simulations, with 4 m/yr being too low for

most of the RCP-8.5 scenario and 16 m/yr being too high for the majority of scenarios and ensemble samples."

*Response: Done.*

Line 37: can you add "(described in appendix A)" after "ice sheet models."?

*Response: Done.*

Line 37 and section 2.5: This paragraph seems out of place as its own section, being relatively short and quite necessary for the discussion at the beginning of section 2.4 I recommend removing section 2.5 altogether, and moving its context/text to wrap into and support the first few sentences of section 2.4.

*Response: This needs to stay for completeness.*

Line 37: There are switches between past and present tense throughout. Can you choose one and be consistent in the manuscript?

*Response: Done.*

Pg. 6, line 3: The start of this sentence could be rewritten for clarity as, "Although these simulations are highly interesting, a full discussion of their results..."

*Response: Done.*

Line 6: remove the redundant "the uncertainty in."

*Response: Done.*

Line 22: please add ", A_mu(t)," after "the observed response" to note the equation terms.

*Response: Done.*

Line 24: This section is a bit difficult to follow. I suggest removing "the response function is obtained" and adding a sentence like, "Following this procedure and using the fixed Heavyside forcing mu=8m/yr chosen above, we obtain the response functions for each of the ice-sheet models."

*Response: Done in a slightly different way.*

Line 38: I suggest rewriting "for the 200 years of the forcing period" as "which is held constant over 200 years."

*Response: Done.*

Pg. 7, line 25-26: "but not that far off in most cases" is rather general. A short but descriptive finish to this sentence would improve and contextualize this paragraph. I suggest rewriting as: "but in most cases the assumption is a reasonable approximation of how basal melting is responding to external forcing."

*Response: Done.*

Pg. 8, line 7: "and" should be "with"

*Response: We would like to keep it as it is.*

Line 24: Increases in the ice-sheet associated with precipitation changes cannot be explicitly or implicitly accounted for in your linear response (this is apparent in the figures, as no changes are below

0). But observed ice-sheet changes over this period in the EAIS are increasing in mass. If they are not accounted for, could that also lead to differences between the estimates here and the observations? You mention this in the conclusions, but can you discuss this drawback and its implications in more detail in this section?

*Response: Done.*

Line 31: By "largest" do you refer to the median values?

*Response: We have added a bit more context here.*

Lines 35-36: This sentence is difficult to follow.

*Response: We have rephrased the sentence.*

Line 38: How are uncertainties distributed across the different regions, and why?

*Response: They are distributed according to Figure 3.*

Pg. 9, line 1: Is there a way to decompose equation 5 at each timestep to determine the relative importance of each component at each timestep?

*Response: This is not possible, unfortunately, because the convolution is non-local in time and therefore at each time step the history of the region matters.*

Line 6: "However" is redundant with your previous statement and I do not think is what you mean. Can you use something like "In any case"

*Response: Done.*

Pg. 11: Are the codes and data used to produce these analyses publicly available? Please provide to the extent possible, in accordance with the ESD data policy:

https://www.earth-system-dynamics.net/about/data_policy.html

*Response: Will do.*

Figure 1 caption: It would be helpful to add the bold symbols for the basal melt rate and sensitivity, as was done for the other terms.

*Response: Done.*

Figure 4b-e captions: "Figure 3a" should be "Figure 4a".

*Response: Done.*

Figure 5a: Please note that the response function are the grey lines.

*Response: Done.*

Tables 2-5: Are these coefficients the alpha_r values? It would be helpful label them in the table with their symbol and describe them in words in the caption.

*Response: Done.*

[revised manuscript text omitted]

---

## Author Comment (AC2) · 14 Oct 2019

Please find attached the final response to both reviewers.

Please also note the supplement to this comment:
https://www.earth-syst-dynam-discuss.net/esd-2019-23/esd-2019-23-AC2-supplement.pdf

2019.

---

## Referee Report (RR1)

In revised paper authors have carefully provided the clarification. The authors have also given a brief background to their previous publication to give the reader a better understanding of the work. The present form looks acceptable for publication in Earth System Dynamics.

Some minor comments:
**Page 3 Line 15:** Please rephrase *"An important if not the most important...*

**Page 3 line 23:** *"... which"* to "..., which"

**Page 4 line 7:** *" Whenever the term .."* change to - The term, Antarctic contribution ... used in this study"

**Page 4 line 13:** Is the forcing data same as in Levermann et al., 2014? if so please refer to it for clarity. What about the initial conditions and the spin up time, are they same?

**Page 11 Line 25:** *".. but some errors may remain in some places"* - unlcear: What type of error and which places?

Please use either spin-up or spinup. If the authors refer to different terms please clarify it.

---

## Referee Report (RR2)

**Daniel Gilford**
**11/25/19**

**Second Review of Projecting Antarctica's contribution to future sea level rise from basal ice-shelf melt using linear response functions of 16 ice sheet models (LARMIP-2) by Levermann et al. [ESSD 10.5194/esd-2019-23]**

I appreciate the authors considering and addressing my previous comments. I only have a couple questions for clarification and some minor grammatical suggestions to improve readability. Otherwise, this paper is acceptable for publication and will be a valuable addition to the literature.

Comments:

Pg. 4, line 3: "taking" I think should be "taken"?

Pg. 4, line 17-41: I realize this was probably written in response to a previous reviewer comment, but it is very repetitive with the next page. Perhaps only the summary list is needed and then each of the following sections can provide greater details of the method?

Pg. 6, line 9: What is meant by "this approach may not be valid for absolute values"? Do you mean if DeltaTg and DeltaTo were replaced with Tg and To? If so, I don't think this phrase is needed.

Pg. 6, line 31: Can you rewrite as "display a wide range of total melt rates over space", as this is what I think you mean by this sentence?

Pg. 10, line 14: Although you go into it below, it would be helpful to note here that overall there appears a bias towards more melting in the models than the observed values (Figure 7).

Pg. 10, line 39: This last sentence is repetitive and not necessary, as you explain it in line 30.

Pg. 14, line 13: Can you cite these "previous studies"?

Pg. 15, line 14: Please note that these are the median values.

Pg. 17, line 22: "provide" should be "provides".

References: It's a small thing that will probably be addressed in copy-editing, but it would be helpful to alphabetize the references.

---

## Author Response (AR2)

**Response to reviewers**

We are very happy that the reviewers consider our manuscript ready for publication. We have corrected all errors provided as a list by the reviewers. The questions which were not just typo corrects are detailed below.

5 In order to facilitate the reading of this document, responses to the reviewers are given in blue and italic compared to the reviewer comments which are given in black without italic font.

**Response to reviewer #1**

Page 4 line 13: Is the forcing data same as in Levermann et al., 2014? if so please refer to it for clarity. What about the initial conditions and the spin up time, are they same?

*Response*: We have added notes that the forcing data is the same and the initial conditions are taken from InitMIP. This was mentioned at other places, but we have added it here again for convenience.

Page 11 Line 25: ".. but some errors may remain in some places" - unlcear: What type of error and which places?

15 *Response*: The places where errors occur are listed in the sentences following this one.

**Response to Daniel Gilford's comments (reviewer #2)**

Pg. 4, line 17-41: I realize this was probably written in response to a previous reviewer comment, but it is very repetitive with the next page. Perhaps only the summary list is needed and then each of thefollowing sections can provide greater details of the method?

Response: This was indeed added in order to fulfil another reviewer's request. We are caught between these two assessments and would like to keep it as it is even if it is somewhat redundant. We hope this is alright.

Pg. 6, line 9: What is meant by "this approach may not be valid for absolute values"? Do you mean if DeltaTg and DeltaTo were replaced with Tg and To? If so, I don't think this phrase is needed.

*Response:* We have changed it accordingly.

Pg. 6, line 31: Can you rewrite as "display a wide range of total melt rates over space", as this is what I think you mean by this sentence?

Response: Done.

30 Pg. 10, line 14: Although you go into it below, it would be helpful to note here that overall there appears a bias towards more melting in the models than the observed values (Figure 7).

Response: Done.

Pg. 10, line 39: This last sentence is repetitive and not necessary, as you explain it in line 30.

Response: We erased it.

**Projecting Antarctica's contribution to future sea level rise from basal ice-shelf melt using linear response functions of 16 ice sheet models (LARMIP-2)**

Anders Levermann1,2,3,\*, Ricarda Winkelmann1,3, Torsten Albrecht1, Heiko Goelzer4,5, Nicholas R.

- 5 Golledge6,7, Ralf Greve8, Philippe Huybrechts9, Jim Jordan10, Gunter Leguy11, Daniel Martin12, Mathieu Morlighem13, Frank Pattyn5, David Pollard14, Aurelien Quiquet15, Christian Rodehacke16,17, Helene Seroussi18, Johannes Sutter17,19, Tong Zhang20, Jonas Van Breedam9, Robert DeConto21, Christophe Dumas15, Julius Garbe1,3, G. Hilmar Gudmundsson10, Matthew J. Hoffman20, Angelika Humbert17,22, Thomas Kleiner17, William H. Lipscomb11, Malte Meinshausen23,1, Esmond Ng12, Sophie M.J.
- 10 Nowicki24, Mauro Perego25, Stephen F. Price20, Fuyuki Saito26, Nicole-Jeanne Schlegel18, Sainan Sun5, Roderik S.W. van de Wal4,27

2LDEO, Columbia University, New York, USA

[revised manuscript text omitted]